# CROSS-DOMAIN CASCADED DEEP TRANSLATION

## ABSTRACT

In recent years we have witnessed tremendous progress in unpaired image-to-image translation methods, propelled by the emergence of DNNs and adversarial training strategies. However, most existing methods focus on transfer of *style* and *appearance*, rather than on *shape* translation. The latter task is challenging, due to its intricate non-local nature, which calls for additional supervision. We mitigate this by descending the deep layers of a pre-trained network, where the deep features contain more semantics, and applying the translation between these deep features. Specifically, we leverage VGG, which is a classification network, pre-trained on ImageNet with large-scale semantic supervision. Our translation is performed in a cascaded, deep-to-shallow, fashion, along the deep feature hierarchy: we first translate between the deepest layers that encode the higher-level semantic content of the image, proceeding to translate the shallower layers, conditioned on the deeper ones. We show that our method is able to translate between different domains, which exhibit significantly different shapes. We evaluate our method both qualitatively and quantitatively and compare it to state-of-the-art image-to-image translation methods. Our code and trained models will be made available.

## 1 INTRODUCTION

In recent years, neural networks have significantly advanced generative image modeling. Following the emergence of Generative Adversarial Networks (GANs) (Goodfellow et al., 2014), image-to-image translation methods have dramatically progressed, revolutionizing applications such as inpainting (Yu et al., 2018), super resolution (Wang et al., 2018), domain adaptation (Hoffman et al., 2017), and more. In particular, there have been intriguing advances in the setting of unpaired image-to-image translation through the use of cycle-consistency (Yi et al., 2017; Zhu et al., 2017) as well as other approaches (Benaim & Wolf, 2017; Huang et al., 2018; Lee et al., 2018; Ma et al., 2018). However, most existing methods acknowledge the difficulty in translating *shapes* from one domain to another, as this might entail highly non-trivial geometric deformations. Consider, for example, translating between elephants and giraffes, where we would expect the neck of an elephant to be extended, while the elephant's head should shrink. The challenge is compounded by the fact that, even within the same domain, images might exhibit extreme variations in object shape and pose, partial occlusions, and contain multiple instances of the object of interest. One might even argue that this translation task is ill-posed to begin with, and at the very least, requires high-level semantics to be accounted for.

Nonetheless, several works do address shape deformation in the context of image-to-image translation by requiring supervision in the form of a foreground mask (Liang et al., 2018; Mo et al., 2018). In contrast, GANimorph (Gokaslan et al., 2018) and the recently proposed TransGaGa (Wu et al., 2019) show remarkable translation without requiring additional supervision for several datasets. However, these techniques excel in controlled setting only, where the images are controlled, and the foreground separation is rather simple.

In this paper, we address the problem of unpaired image-to-image translation, without requiring foreground masks, between two different domains, where the objects of interest share some semantic similarity (e.g., four-legged mammals), whose shapes and appearances may, nevertheless, be drastically different. Our key idea is to accomplish the translation task by learning to translate between deep feature maps. Rather than learning to extract the relevant higher-level semantic information for the specific pair of domains at hand, we leverage deep features extracted by a network pre-trained for image classification, thereby benefiting from its large-scale fully supervised training.

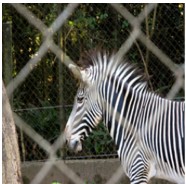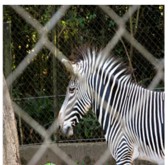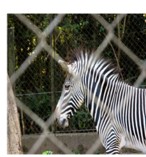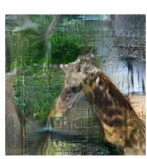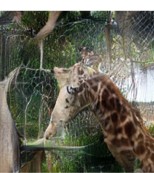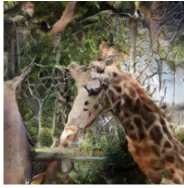

Figure 1: Given an image from domain $A$ (zebras), we extract its deep features using a network pre-trained for classification, specifically VGG-19 pre-trained on ImageNet, and translate them into deep features of domain $B$ (giraffes). We first translate high level semantics encoded in `conv_5_1` of the zebra to those of a giraffe, represented by the inner images. Then, we use a cascade of deep-to-shallow adversarially trained translators, one for each deep feature layer, to translate shallower layers, i.e. `conv_4_1` (middle sized zebra and giraffe) and then `conv_3_1` (outer images) . The presented images were obtained by training feature inversion networks, from deep features to image space.

Our work is motivated by the well-known observation that neurons in the deeper layers of pre-trained classification networks represent larger receptive fields in image space, and encode higher-level semantic content Zeiler & Fergus (2014). In other words, local activation patterns in the deeper layers may encode very different shapes in size and structure. Furthermore, Aberman et al. (2018) showed that semantically similar regions from different domains, e.g. dog and cat, have similar activations. That is, the encoding of a cat's eye resembles that of a dog's eye more than that of its tail. From the point of view of the translation task, these properties are attractive, since they suggest that it might be possible to learn a *semantically consistent* translation between activation patterns produced by images from different domains, and that the resulting (reconstructed) image would be able to change drastically, hopefully bypassing the common difficulties in image-to-image translation methods.

More specifically, we learn to translate between several layers of deep feature maps, extracted from two domains by a pre-trained classification network, namely VGG-19 (Simonyan & Zisserman, 2014). The translation is carried out one layer at a time in a deep-to-shallow (coarse-to-fine) *cascaded* manner. For each layer, we adversarially train a dedicated translator that acts in the features space of that layer. The deepest layer translator effectively learns to translate between semantically similar global structures, such as body shape or head position, as demonstrated by the middle pair of images in Figure 1. The translator of each shallower layer is conditioned on the translation result of the previous layer, and learns to add fine scale and appearance details, such as texture. At every layer, in order to visualize the generated deep features, we use a network pre-trained for inverting the deep features of VGG-19, following the method of Dosovitskiy & Brox (2016). The images shown in Figure 1 were generated in this manner.

Our conceptual novelty, can be viewed as a transfer learning for image translation as we are translating high level semantics, encoded in the deeper layers of a pre-trained classification network, a.k.a deep features. This is in contrast to existing methods Gokaslan et al. (2018); Wu et al. (2019), which learn to translate the images directly. We compare our method with several state-of-the-art image translation methods. To demonstrate the effectiveness of our approach, we present results for several challenging pairs of domains that exhibit drastically different shapes and appearances, but share some high-level semantics. Our translations are semantically consistent, typically preserving pose and number of instances of objects of interest, and reproducing their partial occlusion or cropping, as may be seen in Figure 5.

## 2 RELATED WORK

Zhu et al. (2017) (simultaneously with Yi et al. (2017) and Kim et al. (2017)) have presented remarkable unpaired image-to-image translations, using a framework known as CycleGAN. The key idea is that the ill-posed conditional generative process can be regularized by a cycle-consistency constraint, which enforces the translation to perform a bijective mapping. The cycle constraint has become a popular regularization technique for unpaired image-to-image translation. For example, the UNIT framework (Liu et al., 2017) assumes a shared latent space between the domains and enforces the cycle constraint in the shared latent space. Several works were developed to extend the one-to-one mapping to many-to-many mapping (Ma et al., 2018; Huang et al., 2018; Lee et al., 2018; Almahairi

et al., 2018). These methods decompose the encoding space to shared latent space, representing the domain invariant content space, and domain specific style space. Therefore, many translations can be achieved from a single content code by changing the style code of the input image.

Many translation methods share the inability to translate high-level semantics, including different shape geometry. This type of translation is usually necessary in the case of transfiguration, where one aims to transform a specific type of objects without changing the background. Both Lee et al. (2018) and Mejjati et al. (2018) learn an attention map and apply translation only on the the foreground object. However, both methods only improve translations that do not deform shapes. Gokaslan et al. (2018) succeed in preforming several shape-deforming translations by several modifications to the CycleGAN framework, including using dilated convolutions in the discriminator. However, they have not demonstrated strong shape deformations, such as zebras to elephants or giraffes, as we show in Section 4.

Liang et al. (2018) and Mo et al. (2018) assume some kind of segmentation is given, and use this segmentation to guide shape deformation translation. However, such segmentation is hard to achieve. In a recent work, Wu et al. (2019) disentangle the input images to geometry and appearance spaces, relying on high intra-consistency, and learn to translate each of the two domains separately. However, the variation of geometry and appearance of in-the-wild images is too large to be disentangled successfully[1].

Contrary to the above works, our work leverages a pre-trained network and the translation is applied directly on deep feature maps, thus being guided by high-level semantics. Several image-to-image methods, such as Xu et al. (2018); Di et al. (2018); Ignatov et al. (2018), also incorporate such pre-trained networks, though usually, only as perceptual loss, constraining the translated image to remain semantically close to the input image. Differently, Sungatullina et al. (2018) incorporate pre-trained VGG features into the discriminator architecture, to assist in the discrimination phase. Wu et al. (2018) use VGG-19 as a fixed encoder in the translation, where only the decoder is learned. Upchurch et al. (2017) present the only method, to our knowledge, that actually translates deep features between two domains. However, the translation is not learned, but defined by simply interpolating between the deep features, which restricts the scope of method to highly aligned domains. For completeness, we also mention that Yin et al. (2019) train an autoencoder to embed point clouds, and perform translation in the learned embedding. In contrast, we utilize semantics to preform the translation in the much more difficult scenario of images.

Our work shares some similarities with the early work of Huang et al. (2017), which suggests using a generative adversarial model (Goodfellow et al., 2014) in a coarse-to-fine manner with respect to a pre-trained encoder. The generation process begins from the deepest features and then recursively synthesizes shallower layers conditioned on the deeper layer, until generating the final image. This method was only applied on small encoders and low resolution images and was not explored for very deep and semantic encoding neural networks such as VGG-19 (Simonyan & Zisserman, 2014).

Deep image analogies (Liao et al., 2017) transfer visual attributes between semantically similar images, by feed-forwarding them through a pre-trained network. Their work does not train a generative model; nonetheless, they create new deep features by fusing content features from one image with style features of another. Similarly, Aberman et al. (2018) synthesize hybrid images from two aligned images by selecting the dominant deep features response.

## 3 METHOD

Our general setting is similar to that of previous unpaired image-to-image translation methods. Given images from two domains, $A$ and $B$, our goal is to learn to translate between them. However, unlike other image-to-image translation methods, we perform the translation on the deep features extracted by a pre-trained classification network, specifically VGG-19 (Simonyan & Zisserman, 2014).

The translation is carried out in a deep-to-shallow (coarse-to-fine) manner, using a cascade of pairs of translators, one pair per layer. The entire architecture used to train the translators is shown schematically in Figure 2, while Figure 3 illustrates the test-time translation (inference) process. Once

---

[1]Unfortunately, at the time of this submission the authors of Wu et al. (2019) were unable to release their code or train their network on the datasets presented in our paper.

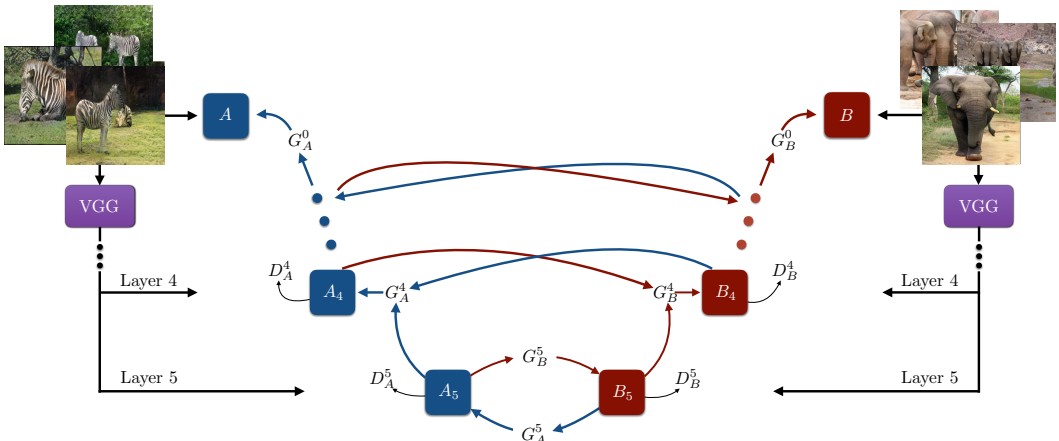

Figure 2: Translation architecture. We translate between domains A and B starting from the deepest feature maps $A_5$ and $B_5$, which encode the highest level semantic content of the images. Translation proceeds from deeper to shallower feature maps until reaching the image itself. The feature maps are extracted by feed forwarding every image through the pre-trained VGG-19 network and sampling five of its layers. Every layer's translation is learned individually, conditioned on the translation result of the next deeper layer (except the deepest layer, whose translation is unconditional).

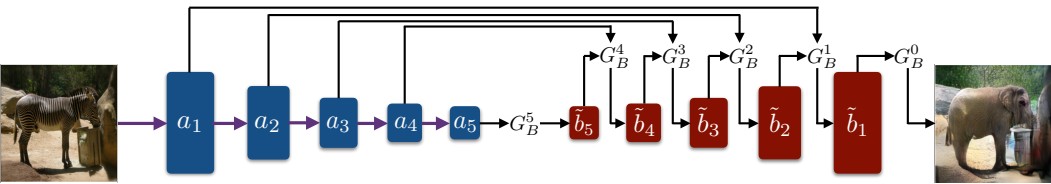

Figure 3: Translation of ~~the top left~~ image at test time. The input image (left) is fed forward through VGG-19 layers, as indicated by the ~~right~~ purple arrows. Then, starting from the deepest layer $a_5$ we translate each layer. The final result is obtained from the shallowest layer using feature inversion.

the deepest feature map has been translated, we translate the next (shallower and less semantic feature map), conditioned on the translated deeper layer. In this manner, the translation of the shallower map preserves the general structure of the translated deeper one, but adds finer details, which are not encoded in the deeper feature maps. We repeat this procedure until the original image level is reached. Below we describe the training and the inference processes in more detail.

**Pre-processing:** We extract high-level semantic features from input images from both domains, $A$ and $B$, by feed-forwarding the images through the pre-trained VGG-19 (Simonyan & Zisserman, 2014) network. Next, we sample five of the resulting deep feature maps, specifically `conv_i_1` (`i` $= 1, 2, 3, 4, 5$), where each map has progressively coarser spatial resolution, but a larger number channels. We denote the $i$-th sampled feature map for image $a \in A$ as $a_i$. Since propagation through the pre-trained VGG-19 network may yield features in any range, while the range of the synthesized features is usually known, to ease the translation, we first normalize each channel, of every layer $i$, by calculating its mean and standard deviation across the domain and then we clamp the normalized feature values to the range of $[-1, 1]$. While the clamping is a potentially harmful irreversible operation, we did not observe any adverse effect on the results. We use $A_i$ ($B_i$) to denote set of all normalized deep feature maps of level $i$, extracted from images in domain $A$ ($B$).

**Inference:** We perform the translation in a coarse-to-fine fashion. Thus, the translator from domain $A$ to $B$, actually consists of a sequence of translators $\left\{G_B^5, G_B^4, \ldots, G_B^1\right\}$, where each translator is responsible for translating the $i$-th feature map layer $a_i$, from $A_i$ to $B_i$ conditioned on the previously translated deeper layer $\tilde{b}_{i+1}$ (except for the deepest layer translator $G_B^5$, which is unconditioned).

Finally, $G_B^0$ uses feature inversion to convert $\tilde{b}_1$ to obtain the translated image. The translators $G_A^i$ from domain $B_i$ to $A_i$ are defined symmetrically. The entire inference pipeline is shown in Figure 3.

**Feature inversion:** In all the results we show, e.g., Figure 1, we visualize the output of the various translators by pre-training a deep feature inversion network (per domain), for each layer $i = 1, \ldots, 5$, following Dosovitskiy & Brox (2016). The network aims to reconstruct the original image given the feature map of a specific layer, regularized by adversarial loss so that the reconstructed image would lie in the manifold of natural images. For more details we refer the reader to Dosovitskiy & Brox (2016). The specific settings used in our implementation are elaborated in Appendix A.1.

**Deepest layer translation:** We begin by translating the deepest feature maps, encoding the highest-level semantic features, i.e., $A_5$ and $B_5$, hence, our problem is reduced to translating high-dimensional tensors. Our solution builds upon the commonly used CycleGAN framework (Zhu et al., 2017). Specifically, we use the three losses proposed in Zhu et al. (2017). First, in order to generate deep features in the appropriate domain, we utilize an adversarial domain loss $\mathcal{L}_{adv}$. We simultaneously train two translators $G_A^5, G_B^5$ which try to fool domain-specific discriminators, $D_A^5, D_B^5$ (for domains $A_5, B_5$, respectively). However, differently from Zhu et al. (2017) and other image translation methods (Huang et al., 2018; Mo et al., 2018), we have found LSGAN (Mao et al., 2017) not to be well-suited for our task, leading to mode collapse or convergence failures. Instead, we found WGAN-GP (Gulrajani et al., 2017) more effective, thus, the adversarial loss for translation from $X$ to $Y$ is defined as

$$\mathcal{L}_{adv}\left(G_Y, D_Y, X, Y\right) = \mathbb{E}_{x \sim \mathbb{P}_X}\left[D_Y\left(G_Y\left(x\right)\right)\right] - \mathbb{E}_{y \sim \mathbb{P}_Y}\left[D_Y\left(y\right)\right] + \lambda_{gp} \mathbb{E}_{\hat{y} \sim \mathbb{P}_{\hat{Y}}}\left[\left(\|\nabla D_Y\left(\hat{y}\right)\| - 1\right)^2\right], \quad (1)$$

where $G_Y : X \to Y$ is the translator, $D_Y$ is the target domain discriminator, $\lambda_{gp} = 10$ in all our experiments, and $\mathbb{P}_{\hat{Y}}$ is defined by uniformly sampling along straight lines between $\tilde{y} \sim G\left(\mathbb{P}_X\right)$ and $y \sim \mathbb{P}_Y$. For more details we refer the reader to Gulrajani et al. (2017).

Second, for regularizing the translation to one-to-one mapping, we add the cycle constraint,

$$\mathcal{L}_{cyc}(G_X, G_Y, X, Y) = \mathbb{E}_{x \sim \mathbb{P}_X} \|G_X\left(G_Y\left(x\right)\right) - x\| + \mathbb{E}_{y \sim \mathbb{P}_Y} \|G_Y\left(G_X\left(y\right)\right) - x\|, \quad (2)$$

where $\|\cdot\|$ stands for the $L_1$ norm.

Finally, as in Zhu et al. (2017), we have found it helpful to use an identity loss, which guides the networks to preserve common high level features,

$$\mathcal{L}_{idty}(G_X, G_Y, X, Y) = \mathbb{E}_{x \sim \mathbb{P}_X} \|G_X\left(x\right) - x\| + \mathbb{E}_{y \sim \mathbb{P}_Y} \|G_Y\left(y\right) - y\|. \quad (3)$$

The entire loss combines these components as follows

$$\begin{aligned} \mathcal{L}^5 = &\mathcal{L}_{adv}\left(G_B^5, D_B^5, A_5, B_5\right) + \mathcal{L}_{adv}\left(G_A^5, D_A^5, B_5, A_5\right) \\ &+ \lambda_{cyc}\mathcal{L}_{cyc}(G_A^5, G_B^5, A_5, B_5) + \lambda_{idty}\mathcal{L}_{idty}(G_A^5, G_B^5, A_5, B_5), \end{aligned} \quad (4)$$

where $\lambda_{cyc}$ and $\lambda_{idty}$ were set to 100 in all our experiments.

**Coarse to fine conditional translation:** Consider two successive layers, $a_i \in A_i$ and $a_{i+1} \in A_{i+1}$, where the latter has already been translated, yielding $\tilde{b}_{i+1}$ as the translation outcome (see Figure 3). We next perform the translation of the layer $a_i$ to yield $\tilde{b}_i$, using the translator $G_B^i$, conditioned on $\tilde{b}^{i+1}$. Note that $G_B^i$ is effectively a function of all the previously translated layers.

The architecture of $G_B^i$ is schematically shown in Figure 4. Since shallower layers encode less of the semantic content of the image, it is more difficult to learn how they should be deformed, and thus they are used to transfer "style", while the "content" comes from the already translated deeper layer. Inspired by Huang et al. (2018), we add an adaptive instance normalization (AdaIN) (Huang & Belongie, 2017) component, whose parameters are learned from the current layer. Thus, several layers of $G_B^i$ are normalized according to the AdaIN component. $G_A^i$, which is designed symmetrically, is learned simultaneously with $G_B^i$, as shown in Fig.4(a).

The loss for training these shallower translators is roughly the same as that used for training the deepest translation: it consists of adversarial domain loss, cycle constraint loss, and identity loss.

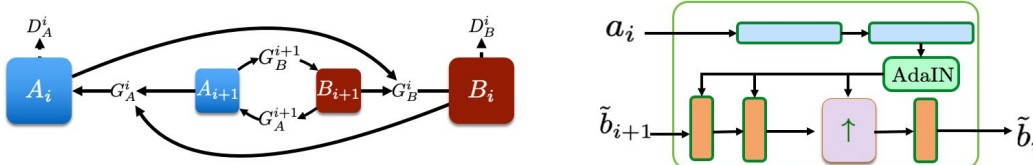

Figure 4: Translation of layer $i$ is conditioned on the previously translated layer $i + 1$. The two translators $G_A^i$ and $G_B^i$ are trained simultaneously (see left figure), where $i + 1, \ldots, 5$ translators are fixed. On the right we show the schematic architecture of $G_B^i$ which has two inputs: $a_i \in A_i$ and $\tilde{b}_{i+1}$. $a_i$ is fed-forward through several layers to yield AdaIN parameters which control the generation of $\tilde{b}_i$. Since $\tilde{b}_i$ has twice the spatial size of $\tilde{b}_{i+1}$, we add an upsampling layer marked by $\uparrow$.

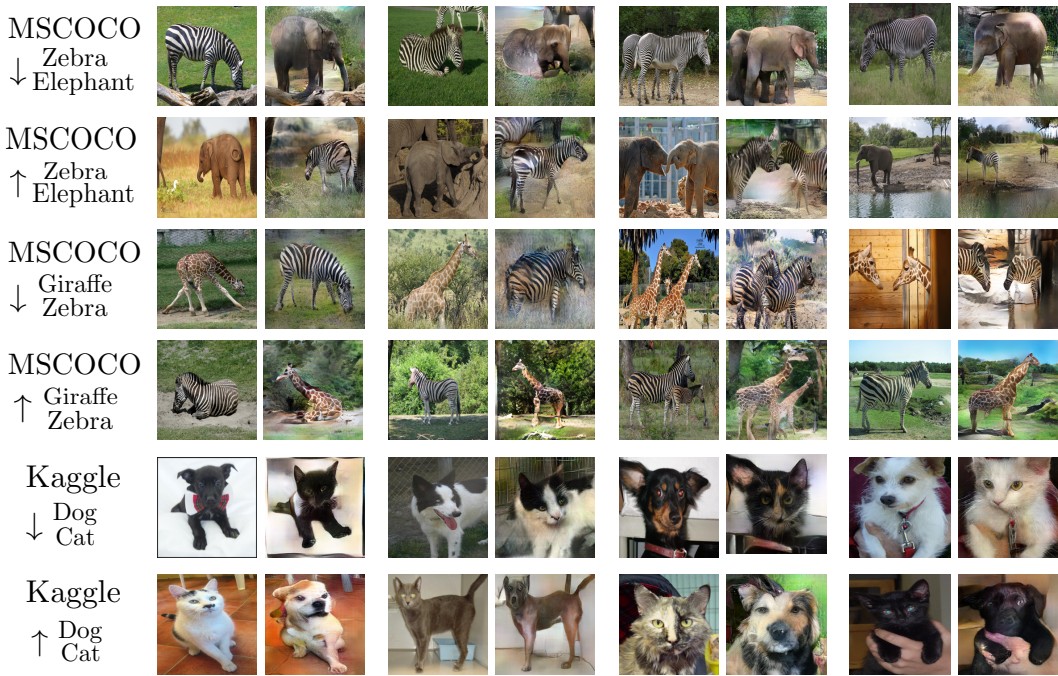

Figure 5: Examples of challenging translation results, featuring significant shape deformations.

While we formalize the adversarial loss unconditionally, similarly to equation 1, the cyclic loss is now conditioned: $\left\| G_A^i \left( G_B^i \left( a_i, \tilde{b}_{i+1} \right), a_{i+1} \right) - a_i \right\| + \left\| G_B^i \left( G_A^i \left( b_i, \tilde{a}_{i+1} \right), b_{i+1} \right) - b_i \right\|$, and the same is true for the identity loss $\left\| G_A^i \left( a_i, a_{i+1} \right) - a_i \right\| + \left\| G_B^i \left( b_i, b_{i+1} \right) - b_i \right\|$.

We train the pairs of translators one layer at a time, starting from $G_A^5$ and $G_B^5$. More details regarding the implementation and the training of the translators are included in Appendix A.1.

## 4 EXPERIMENTS

We evaluate our methods on several publicly available datasets: (1) Cat $\leftrightarrow$ dog faces (Lee et al., 2018), which contains 871 cats images and 1364 dogs images and does not require high shape deformation; (2) Kaggle Cat $\leftrightarrow$ dog (Elson et al., 2007) dataset with over $12,500$ images in each domain (the images may contain part of the object or several instances); (3) Challenging MSCOCO dataset (Lin et al., 2014), specifically, zebra $\leftrightarrow$ elephant and zebra $\leftrightarrow$ giraffe ~~(the number of images is each category is reported in~~ (overall there are 1917 zebras, 2547 giraffes and 2144 elephants). We note that no previous method has used MSCOCO, without supervision in the form of segmentation.

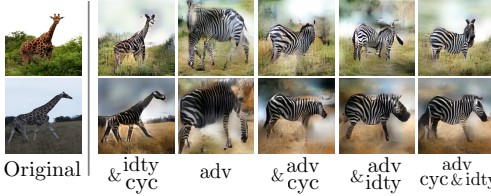

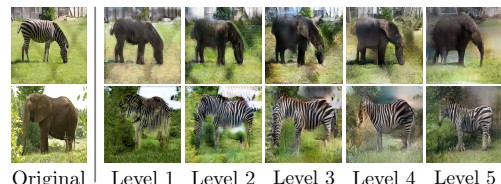

Figure 6: Translation of the 5th (deepest) layer with different loss combinations. Using all three components yields the best result.

Figure 7: Translation of different VGG layers, separately. Low level semantics translation fails to deform the geometry of the object.

| $\rightarrow/\leftarrow$ | Layer 5 | Layer 4 | Layer 3 | Cascaded (ours) |
|---|---|---|---|---|
| Cat $\leftrightarrow$ Dog | 126.93/127.53 | 181.90/164.42 | 178.13/91.71 | **67.58/46.02** |
| Zebra $\leftrightarrow$ Giraffe | 167.62/184.37 | 103.41/53.36 | 112.43 /68.62 | **67.41/39.38** |
| Zebra $\leftrightarrow$ Elephant | 101.26/76 | 105.58/57.34 | 166.32/113.28 | **68.45/47.86** |

Table 1: FID score comparison of different layer translation. Each translation was trained independently. We compare the FID scores on three datasets, measured both directions per dataset. The two directions appear side-by-side, $\rightarrow/\leftarrow$, at each cell.

Our deepest translators, i.e., $G_A^5, G_B^5$, consist of encoder-decoder structure with several strided convolutional layers followed by symmetric transpose convolutional layers. We use group normalization (Wu & He, 2018) and ReLU activation function (except the last layer, which is `tanh`). The conditional generators, consist of learned AdaIN layer, achieved by several strided convolutional layers followed by fully connected layers. The content generator has also several convolutional layers and one single transpose convolutional layer which doubles the spatial resolution (Figure 4(right)). In practice we only train $G^5$, $G^4$, $G^3$, and apply feature inversion directly on the output of the latter, with negligible degradation. For the exact layer's specifics we refer the reader to Appendix A.1 and to our (soon to be published) code. We run each layer for 400 epochs with fixed learning rate of 0.0001 and Adam optimizer (Kingma & Ba, 2014). On a single RTX 2080 reaching the final image takes around 2.5 days, including the final inversion network training.

Several translation examples are presented in Figure 5. Our translation is able to achieve high shape deformation. Note that our translations are semantically consistent, in the sense that they preserve the pose of the object of interest, as well the number of instances is mostly preserved. Furthermore, partial occlusions of such objects, or their cropping by the image boundaries are correctly reproduced. See for example, the translations of the pairs of animals in columns 5–6. More results are provided in Appendix A.2.

### 4.1 ABLATION STUDY

We analyze ~~two~~ main elements of our method.

**Loss components** First, we validate the use of CycleGAN loss components. As shown in Figure 6, we translate the 5th (deepest) layer with and without cycle, identity and adversarial losses. The best approach is achieved by using all of the losses, which balance each other.

**VGG layers** ~~In addition,~~ In Figure 7 we compare translation of different VGG-19 layers. Evidently, shallower layers introduce spatial constraints, thus, limiting the translation in the sense of shape's changes. The shallowest layer can hardly change the shape of the input image, which may explain the failure of traditional image translation methods. In Table 1, we use the common FID score (Szegedy et al., 2015) to show that the cascaded translation achieves better translation compared to individual layer translation. Additional results are shown in Appendix A.4

**Type of pre-trained network** While our method is conceptually agnostic to the type of features extraction network, we still require the extracted features to represent high-level semantic. Therefore, VGG-19 layers, which are the most common features used for image generation tasks (Aberman et al.

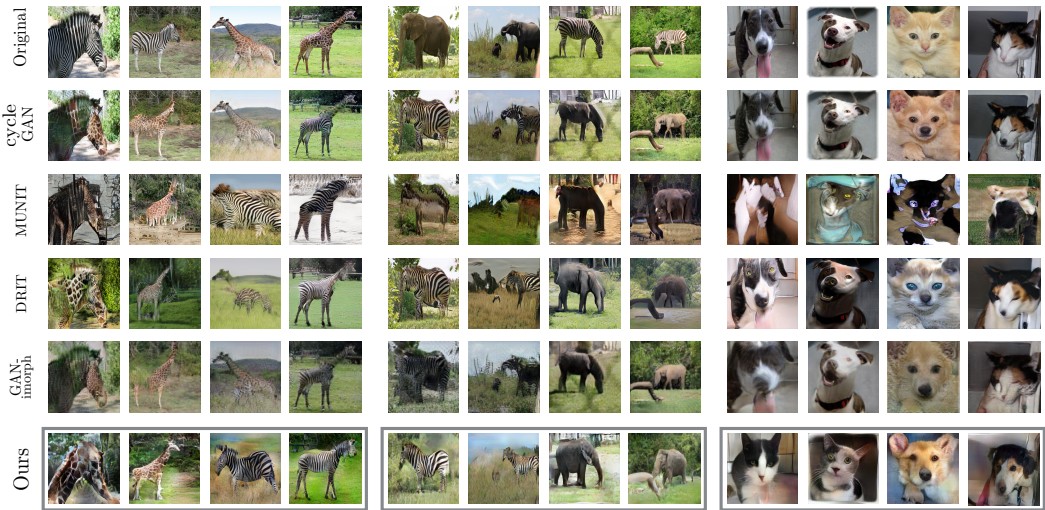

Figure 8: Comparison to other image-to-image translation methods. The unpaired translations, from left to right, are zebra ↔ giraffe, elephant ↔ zebra and dog ↔ cat, where every translation has four examples, two in each direction. While previous translation methods struggle to deform the geometry of the source images, our method is able to preform drastic geometric deformation, while preserving the poses of the subjects and the overall composition of the image.

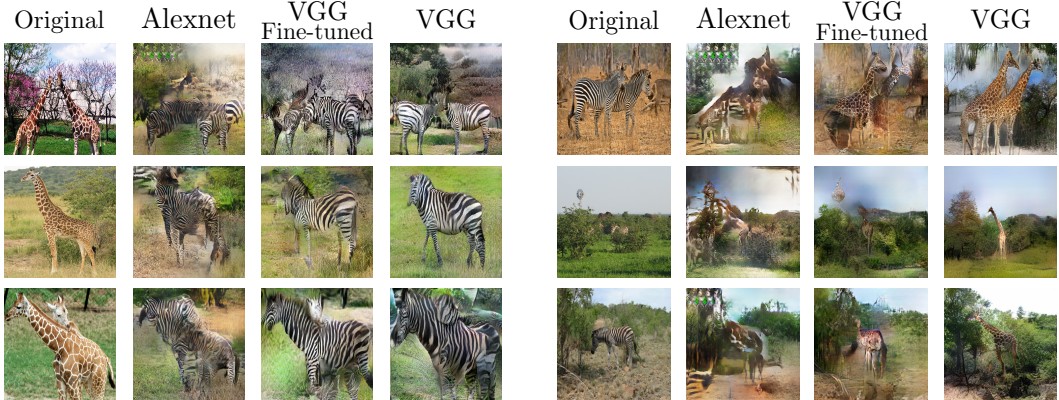

Figure 9: Translation with different pre-trained networks. All the networks were pre-trained on ImageNet. VGG fine-tuned was further trained to classify between zebra and giraffe. Evidently, using fine-tuned VGG-19 does not assist the translation process. In addition, translation of AlexNet features doesn't preserve the shape of the input image.

(2018); Dosovitskiy & Brox (2016); Gatys et al. (2016); Liao et al. (2017)), was chosen. Nonetheless, we experiment with a fine-tuned version of VGG-19 and different network architecture as shown in Figure 9. We fine-tuned VGG to classify between zebras and giraffes and trained our translation networks using the resulting features. As can be seen, the translation results are inferior to the results achieved by the standard VGG-19. This maybe attributed to VGG-19 fixating on the unique differences between zebras and giraffes, unrelated to the translation, such as background. For more about the extracted features, we refer the reader to Appendix A.1.3. In addition, in Figure 9, we examine a different network, AlexNet, also pretrained on ImageNet. We have observed that the deepest image translation was not able to generate a valid shape of zebra or giraffe. AlexNet is composed of 5 convolutional layers, where the first one uses a stride of 4. Therefore, the resulting features have less spatial encoding, especially at the deeper layers, which may explain the difficulty to invert and translate these features.

| →/← | Cycle GAN | MUNIT | DRIT | GANimorph | Ours |
|---|---|---|---|---|---|
| Cat ↔ Dog | 125.75/94.27 | 159.57/108.51 | 153.94/139.17 | 139.17/134.14 | **67.58/46.02** |
| Zebra ↔ Giraffe | **55.65**/58.93 | 238.06/60.78 | 59.75/54.06 | 98.25/120.05 | 67.41/**39.38** |
| Zebra ↔ Elephant | 86.55/68.44 | 109.56/80.1 | 78.01/56.39 | 99.98/89.74 | **68.45/47.86** |

Table 2: FID score comparison. We compare FID scores on three datasets, measured for both translation directions per dataset. The two directions appear side-by-side, →/←, at each cell.

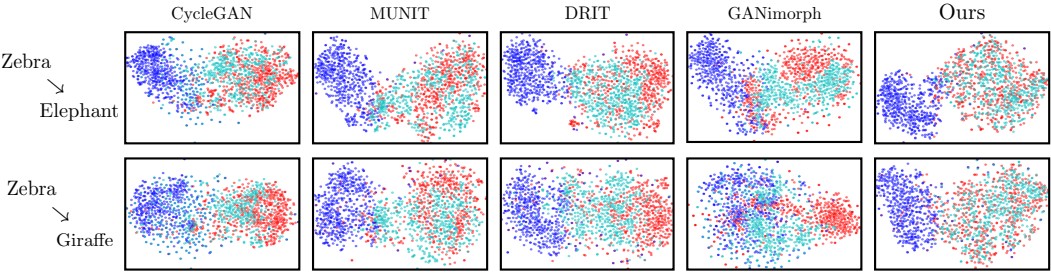

Figure 10: Comparison of the deepest latent spaces (5th layer), projected using t-SNE. The latent space of the source domain is in blue, and the target domain is in red. The distribution of the translation results (in cyan) is most similar to that of the target domain when using our method.

## 4.2 COMPARISON TO OTHER METHODS

We compare our result with leading image translation methods, i.e. CycleGAN (Zhu et al., 2017), MUNIT (Huang et al., 2018), DRIT (Lee et al., 2018) and GANimorph (Gokaslan et al., 2018).

**Quantitative comparison:** In order to perform a quantitative comparison, we use the ~~common~~ FID score (Szegedy et al., 2015), as reported in Table 2. Our method achieved the best FID score on five out of the six cross-domain translations for which this score was measured.

**Qualitative comparison:** In Figure 8 we show several challenging translation examples. While traditional image translation methods struggle to preform translations with such drastic shape deformation, our method is able to do so thanks to its use of the pre-trained VGG-19 network.

The success of our method can also be explained and visualized by examining the translated deep features. We feed forward every image, original and translated, through the entire VGG network, extracting the last fully-connected layer (before the classification layer). We project this vector (of size 4096) to 2D, using t-SNE, as shown in Figure 9. It may be seen that the distribution of the translated vectors (in cyan) is closest to that of the target domain (in red) when using our method.

**Limitations** Our method achieves translations with significant shape deformation in many previously unattainable scenarios, yet, a few limitations remain. First, the background of the object is not preserved, as the background is encoded in the deep features along with the semantic parts. Also, in some cases the translated deep features may be missing small instances or parts of the object. This may be attributed to the fact that VGG-19 is generally not invertible and was trained to classify finite set of classes. In addition, since we translate deep features, small errors in the deep translation may be amplified to large errors in the image, while for image-to-image translation method that operate on the image directly, small translation errors would typically be more local. Please note that, similarly to CycleGAN and GANimorph, our translation is deterministic.

## 5 CONCLUSIONS

Translating between image domains that differ not only changes in appearance, but also exhibit significant geometric deformations, is a highly challenging task. We have presented a novel image-to-image translation scheme that operates directly on pre-trained deep features, where local activation

patterns provide a rich semantic encoding of large image regions. Thus, translating between such patterns is capable of generating significant, yet semantically consistent, shape deformations. In a sense, this solution may be thought of as transfer learning, since we make use of features that were trained for a classification task for an unpaired translation task. In the future, we would like to continue exploring the applications of powerful pre-trained deep features for other challenging tasks, possibly in different domains, such as videos, sketches or 3D shapes.

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

# A APPENDIX

## A.1 TRAINING DETAILS

### A.1.1 Hyper parameters

In all our experiments, unless stated otherwise, we use Adam optimizer [16] with $\beta_1 = 0.5, \beta_2 = 0.999$. The learning rate was set to $0.0001$ and the batch size to $10$. During training, random crop and image mirroring is applied. Our training methodology follows WGAN-GP (Gulrajani et al., 2017), thus for one generator update we update the discriminator four times.

### A.1.2 NETWORK ARCHITECTURE

**Feature Inversion** Our implementation is similar to Dosovitskiy & Brox (2016). We train an individual feature inversion network for VGG layer, where each layer has different channels (512, 512, 256, 128, 64). All layers utilize Leaky ReLU nonlinearity (0.2) and employ no normalization. The last layer utilizes Tanh. All inversion networks, first apply three non-strided convolutional layers, with $N$ input number of channels, equal to the number of channels each deep layer has. Next, several transpose convolutional layers are applied, each doubles the resolution of the image and decreases the channel resolution (by factor of 2) until the image resolution 224 is achieved (thus, different amount of ConvTranspose layers per layer). The final layer is a non-strided convolutional layer followed by Tanh layer. Together they project the features back to the original image dimensions and range (number of output channels is 3). For the discriminator we have used Patch GAN discriminator, with four strided convolutional layers, each utilizes batch normalization (except the first one) and Leaky ReLU. For the adversarial metric, only here, we have used LS-GAN. Here we also set the batch size to 25.

**Deepest layer translation** The input to the deepest translation network is `conv_5_1`, thus, the input size is $14 \times 14 \times 512$ (recall the input image size is $224 \times 224$). The identity and cycle losses are multiplied by $\lambda_{idty} = \lambda_{cyc} = 100$. The architecture is reported in Table.3. The networks is relatively small and achieve good results in a few hours on a single GPU (RTX2080). We use the WGAN-GP optimization method, updating the generator once for every four discriminator updates.

| Name | Input ch. | Output ch. | Kernel sz. | Stride | GN |
|------|-----------|-----------|-----------|--------|-----|
| conv | 512 | 512 | 3 | 1 | no |
| conv | 512 | 256 | 3 | 2 | yes |
| relu | - | - | - | - | - |
| conv | 256 | 512 | 3 | 2 | yes |
| relu | - | - | - | - | - |
| convT | 512 | 256 | 3 | 2 | yes |
| relu | - | - | - | - | - |
| convT | 256 | 256 | 3 | 2 | yes |
| relu | - | - | - | - | - |
| conv | 256 | 512 | 3 | 1 | yes |
| relu | - | - | - | - | - |
| conv | 512 | 512 | 3 | 1 | no |
| tanh | - | - | - | - | - |

Table 3: Deepest layer translation architecture.

**Coarse to fine conditional translation** The coarse-to-fine generator, for generating level $i$, has two inputs: the current source VGG level and the previous translated VGG features $(i + 1)$. An AdaIN component, acts on on the current deep features and normalizes several layers in the translator

itself. We report the AdaIN component structure for generating layer four in Table.4. The architecture can be extended easily to other layers. The core components of the translator, which takes as input the previous translated layer, are reported in Table.5.

| Name | Input ch. | Output ch. | Kernel sz. | Stride | GN |
|---|---|---|---|---|---|
| conv | 512 | 512 | 3 | 2 | no |
| lrelu | - | - | - | - | - |
| conv | 512 | 512 | 3 | 2 | no |
| lrelu | - | - | - | - | - |
| conv | 512 | 512 | 3 | 2 | no |
| lrelu | - | - | - | - | - |
| linear | $4 \times 4 \times 512$ | 1000 | - | - | no |
| lrelu | - | - | - | - | |
| linear | 1000 | $x$ | - | - | no |

Table 4: AdaIN component for the second deepest layer. The output $x$ is equal to the number of parameters the AdaIN normalizes. AdaIN for different VGG layer's translation are defined similarly, where we simply add more conv layer for each shallower VGG layer.

| Name | Input ch. | Output ch. | Kernel sz. | Stride | AdaIN |
|---|---|---|---|---|---|
| conv | $x$ | $x$ | 3 | 1 | yes |
| lrelu | - | - | - | - | - |
| conv | $x$ | $x$ | 3 | 1 | yes |
| lrelu | - | - | - | - | - |
| convT | $x$ | $x/2$ | 4 | 2 | yes |
| lrelu | - | - | - | - | - |
| conv | $x/2$ | $x/2$ | 3 | 1 | no |
| tanh | - | - | - | - | - |

Table 5: Coarse to fine translator. The input number of channels, $x$, varies according to the current VGG layer.

### A.1.3 DIFFERENT NETWORKS

**VGG-19 fine-tuned** We fine-tuned VGG-19 by fixing all layers but the fully connected layers and conv_5_i ($i = 1, 2, 3, 4$). We replaced the final fully connected layer with new fully connected layer of size $4096 \times 2$ corresponding to the two domains. We trained the classifier for 50 epochs, with batch size of 25 and learning rate of 0.01.

**AlexNet** We extract each of the 5 convolution of AlexNet as different layers. Each layer was normalized, in a similar manner as was described for VGG-19.

## A.2 MORE COMPARISON RESULTS

In this section we show more results, not presented in the paper, for zebra↔giraffe, zebra↔elephant and cat↔dog translations.

| Original | Cycle GAN | MUNIT | DRIT | GANimorph | Ours |

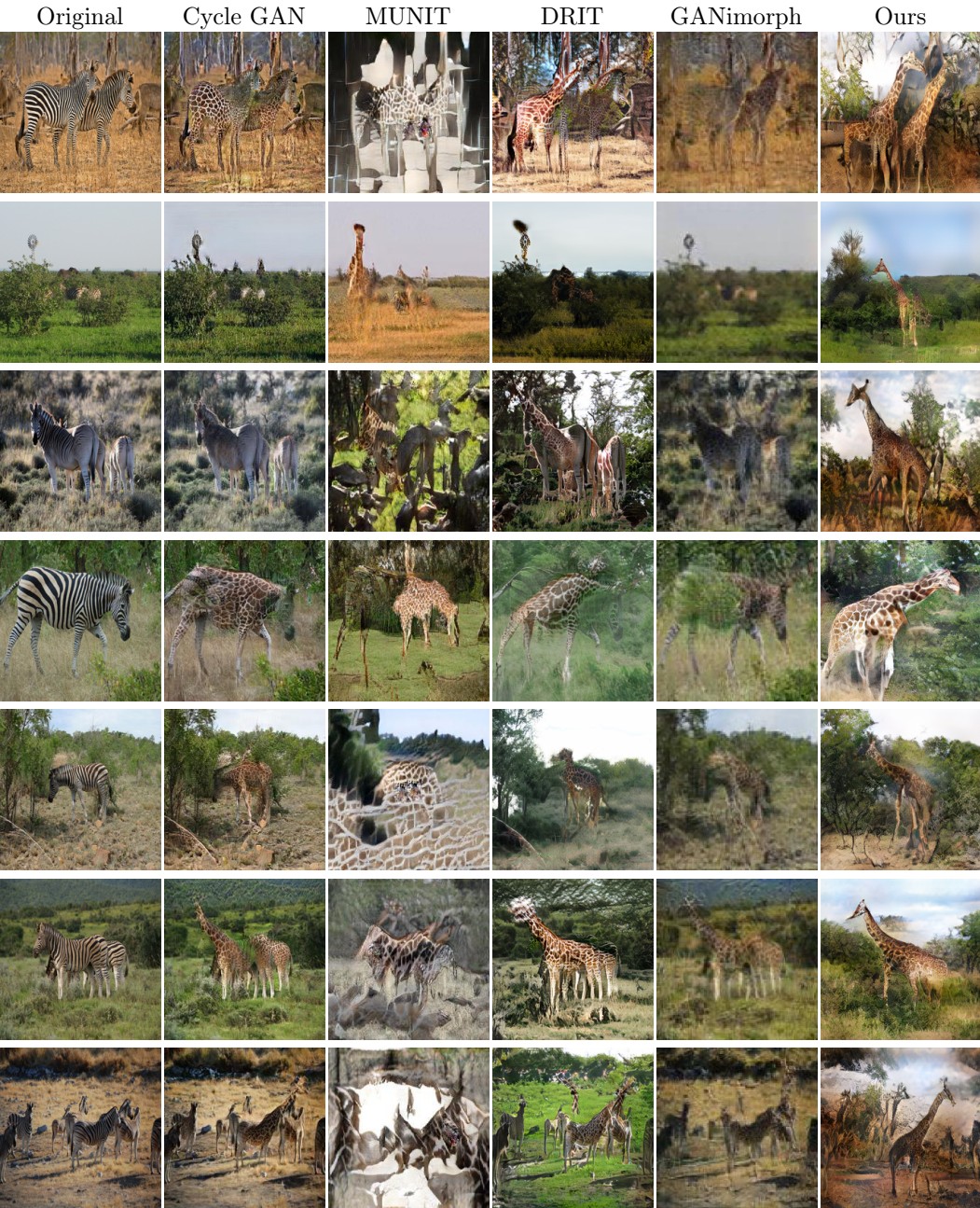

Figure 11: Qualitative comparisons. MSCOCO zebra to giraffe.

Original       Cycle GAN       MUNIT       DRIT       GANimorph       Ours

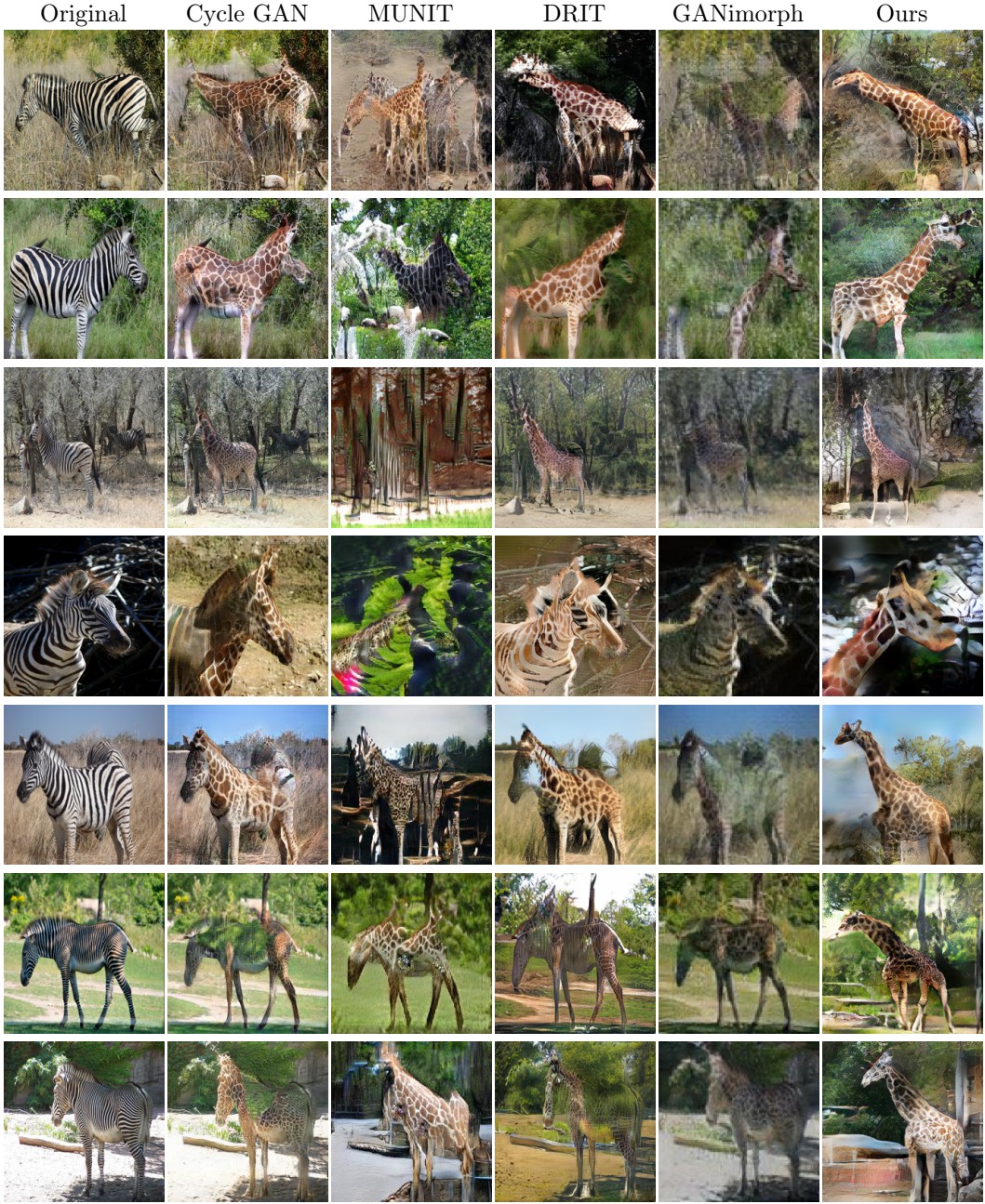

Figure 12: Qualitative comparisons. MSCOCO zebra to giraffe.

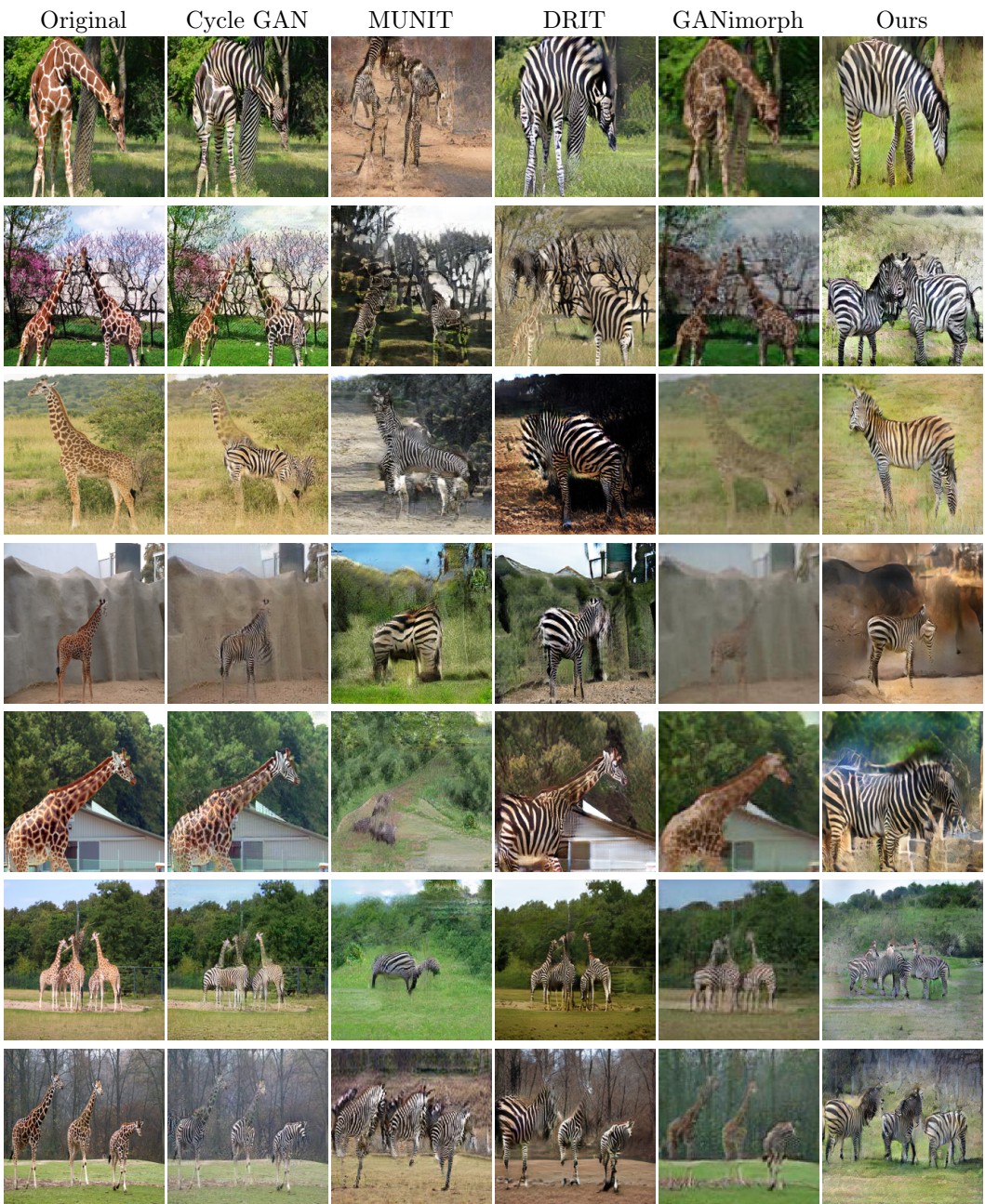

Figure 13: Qualitative comparisons. MSCOCO giraffe to zebra.

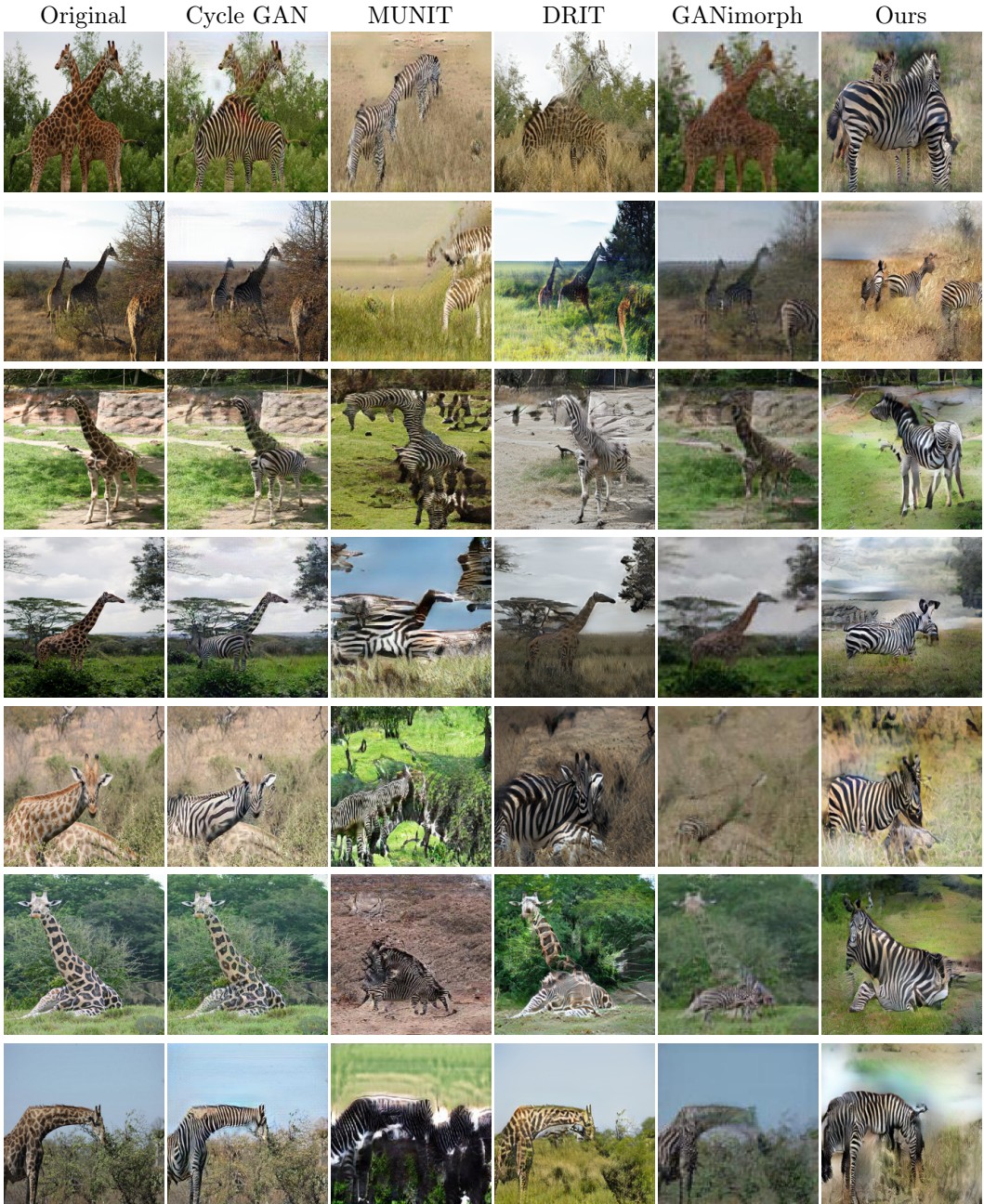

Figure 14: Qualitative comparisons. MSCOCO giraffe to zebra.

Original    Cycle GAN    MUNIT    DRIT    GANimorph    Ours

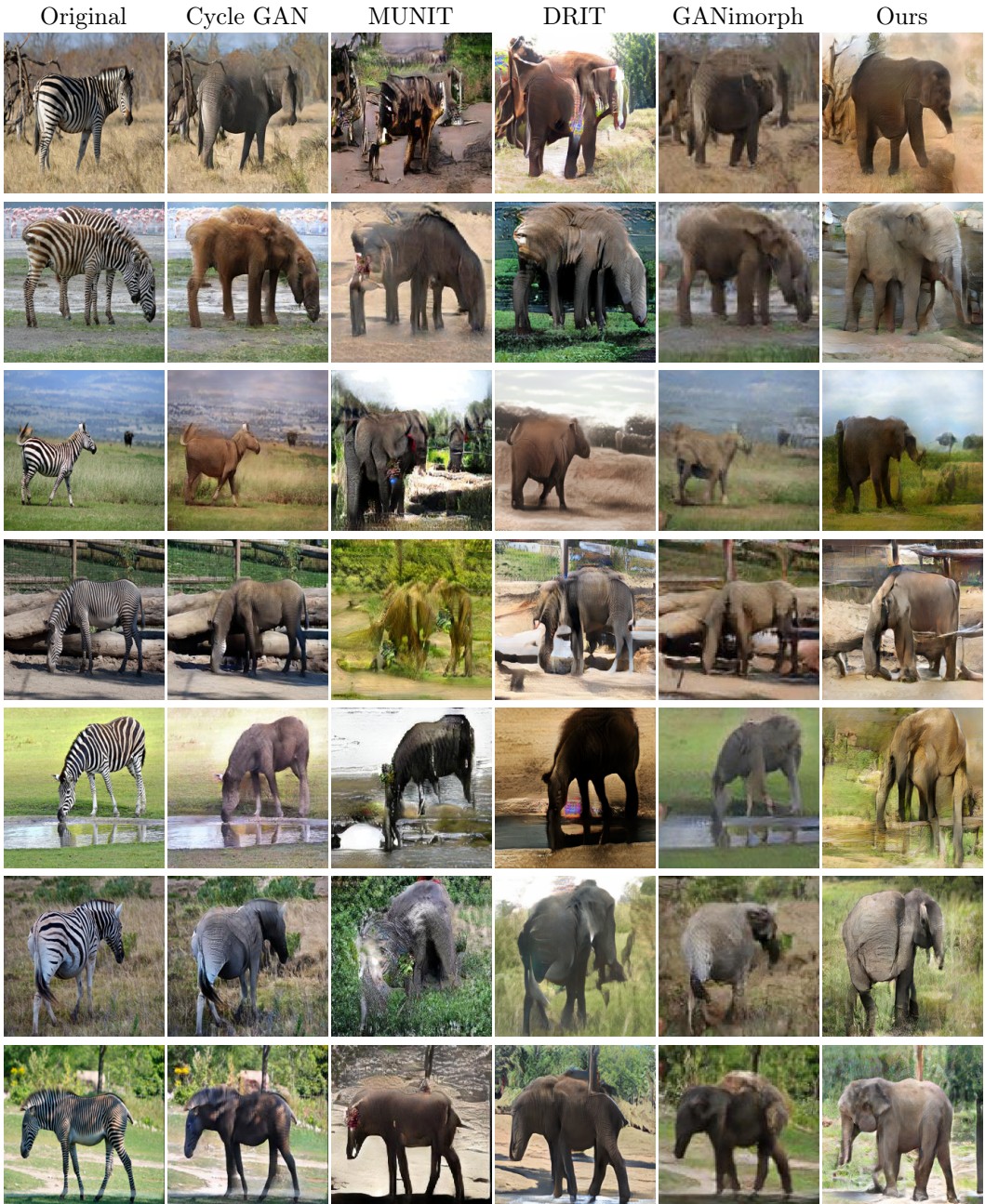

Figure 15: Qualitative comparisons. zebra to elephant.

Original     Cycle GAN     MUNIT     DRIT     GANimorph     Ours

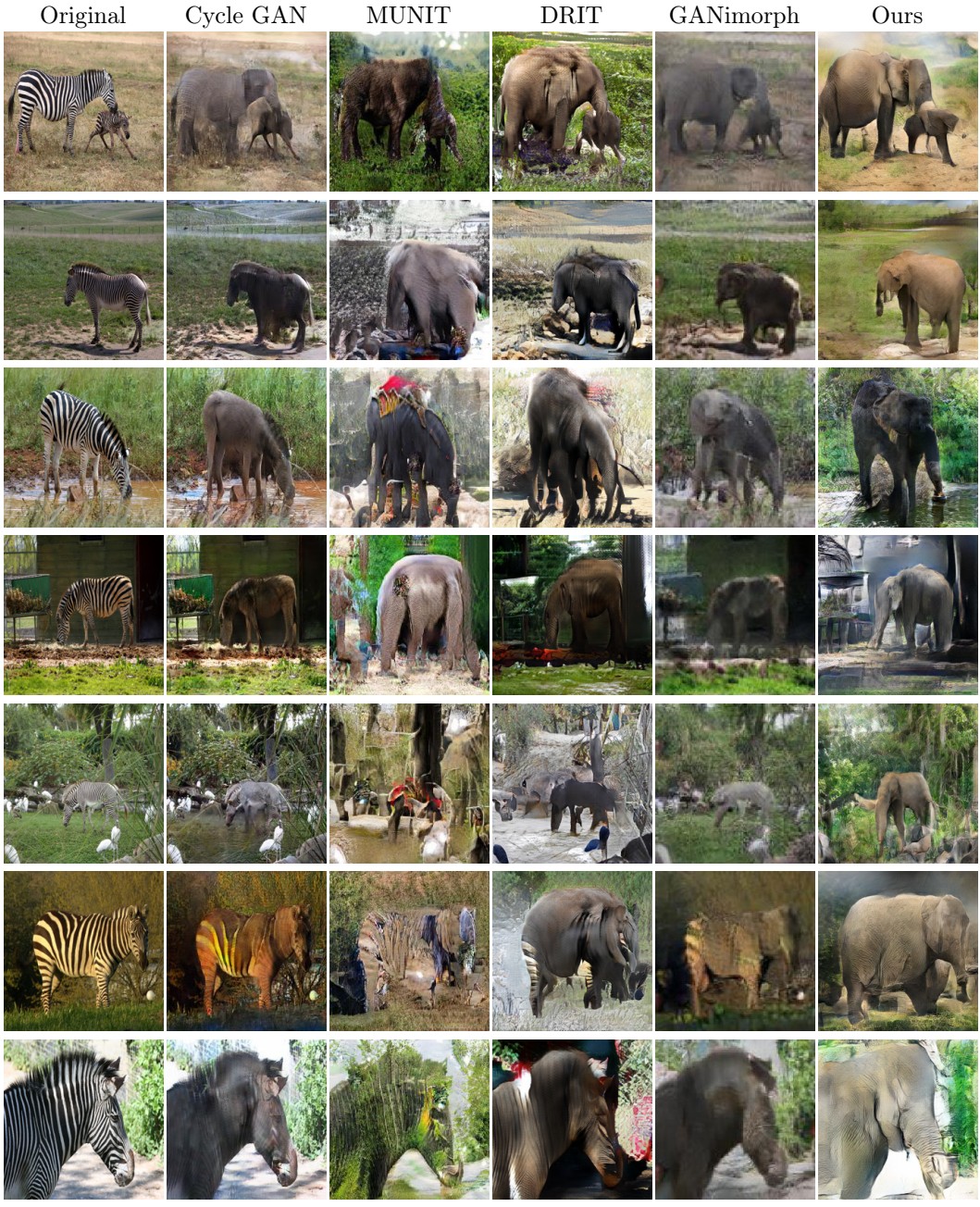

Figure 16: Qualitative comparisons. zebra to elephant.

Original    Cycle GAN    MUNIT    DRIT    GANimorph    Ours

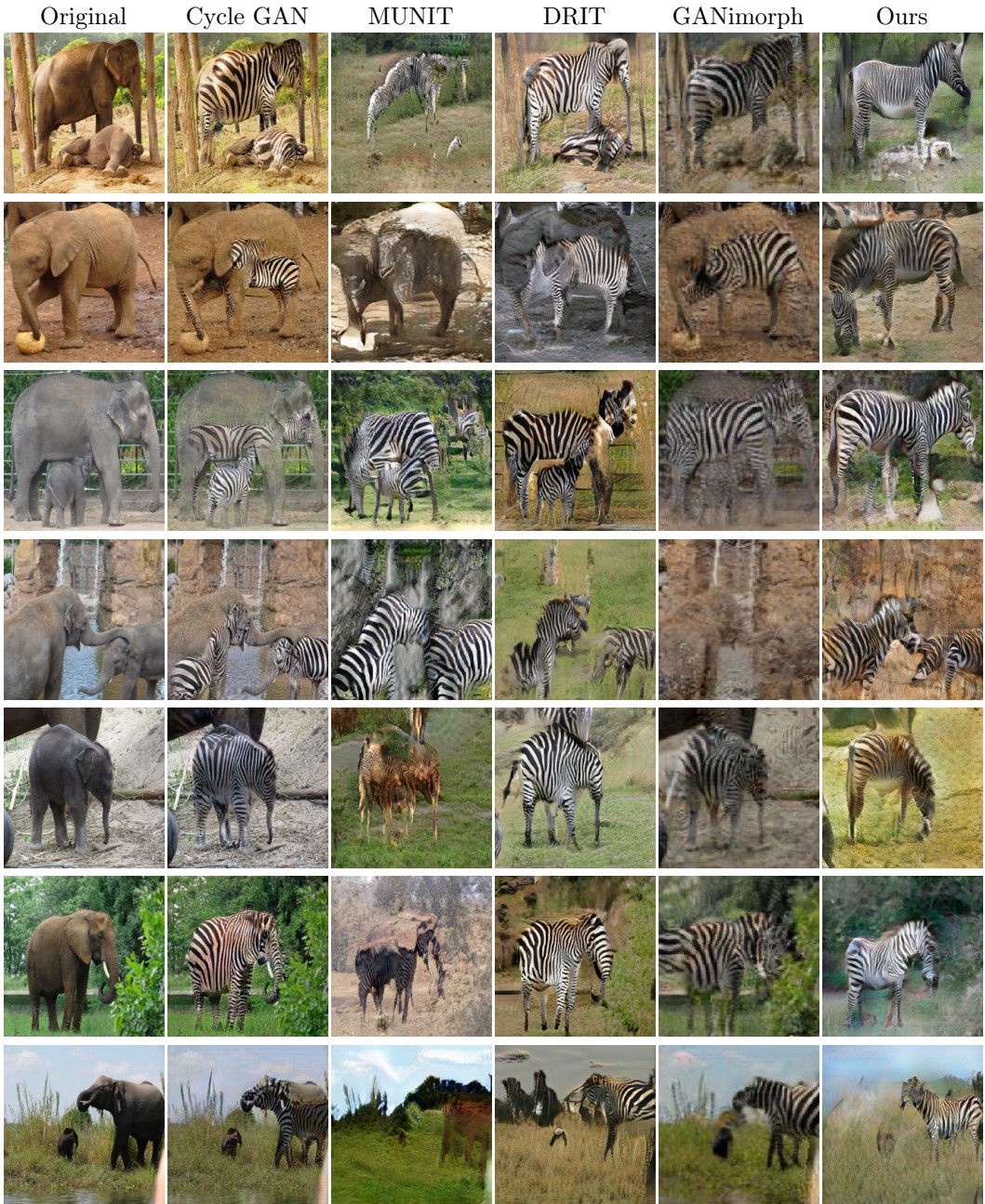

Figure 17: Qualitative comparisons. MSCOCO elephant to zebra.

Original    Cycle GAN    MUNIT    DRIT    GANimorph    Ours

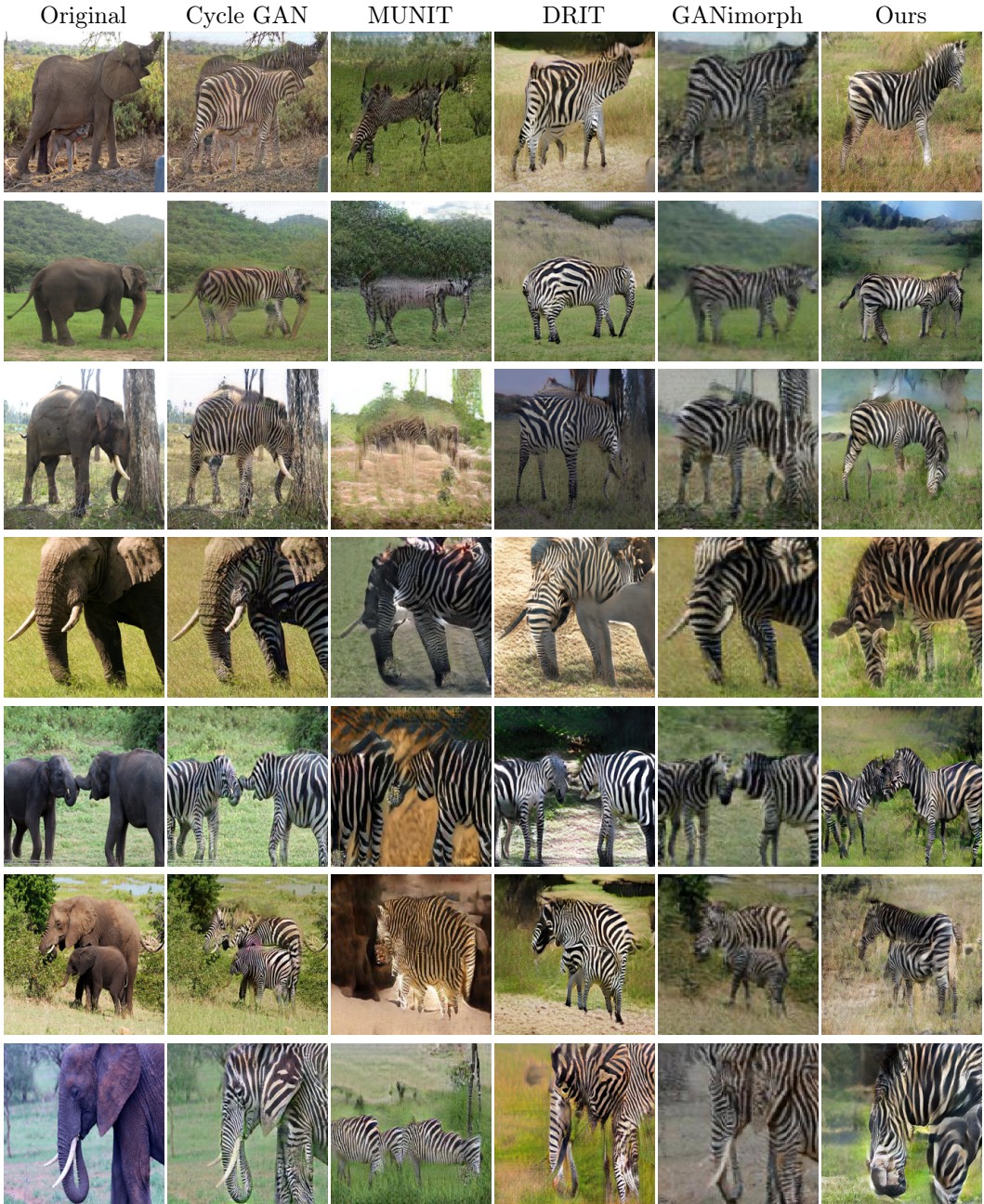

Figure 18: Qualitative comparisons. MSCOCO elephant to zebra.

Original    Cycle GAN    MUNIT    DRIT    GANimorph    Ours

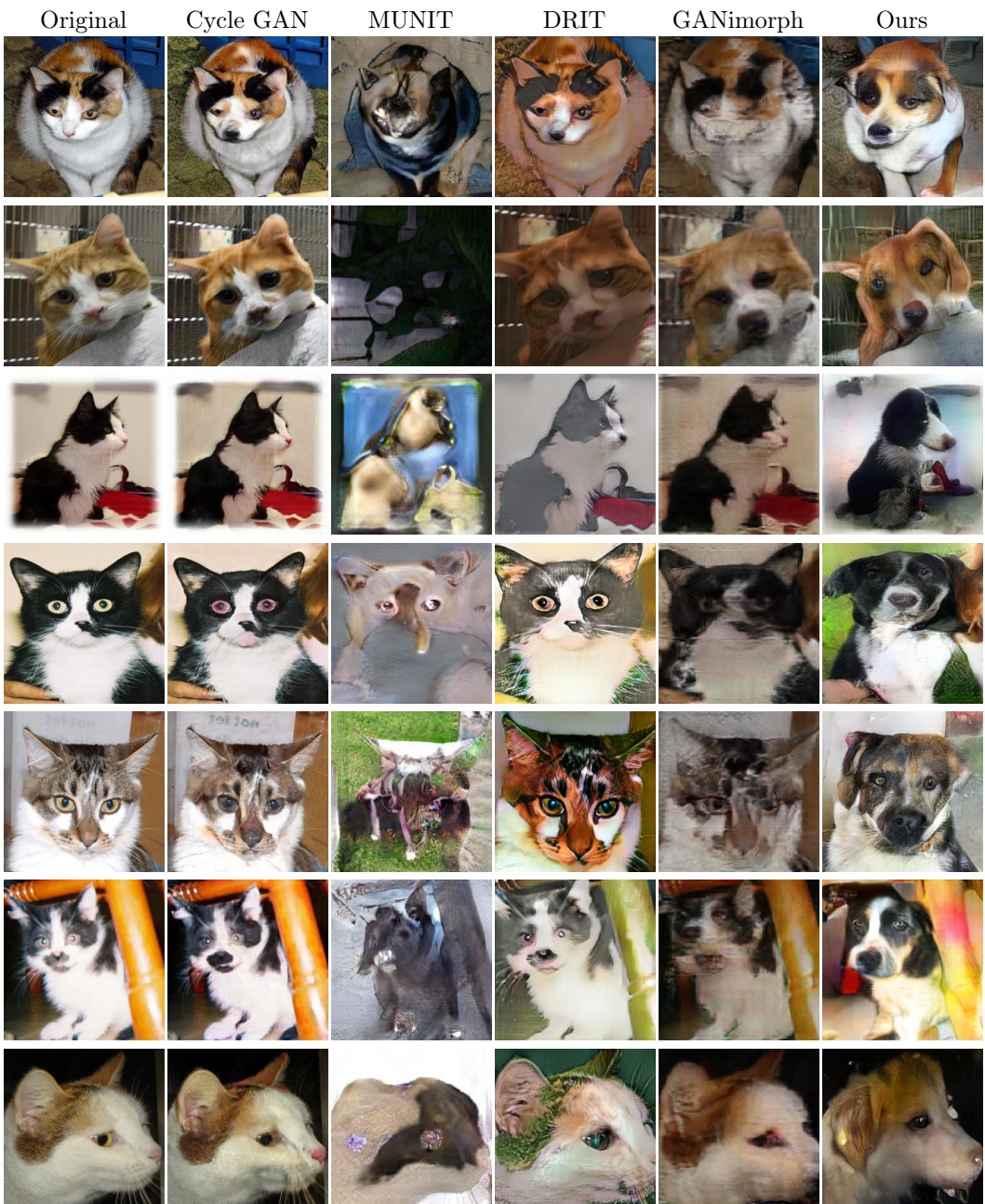

Figure 19: Qualitative comparisons. Kaggle cat to dog.

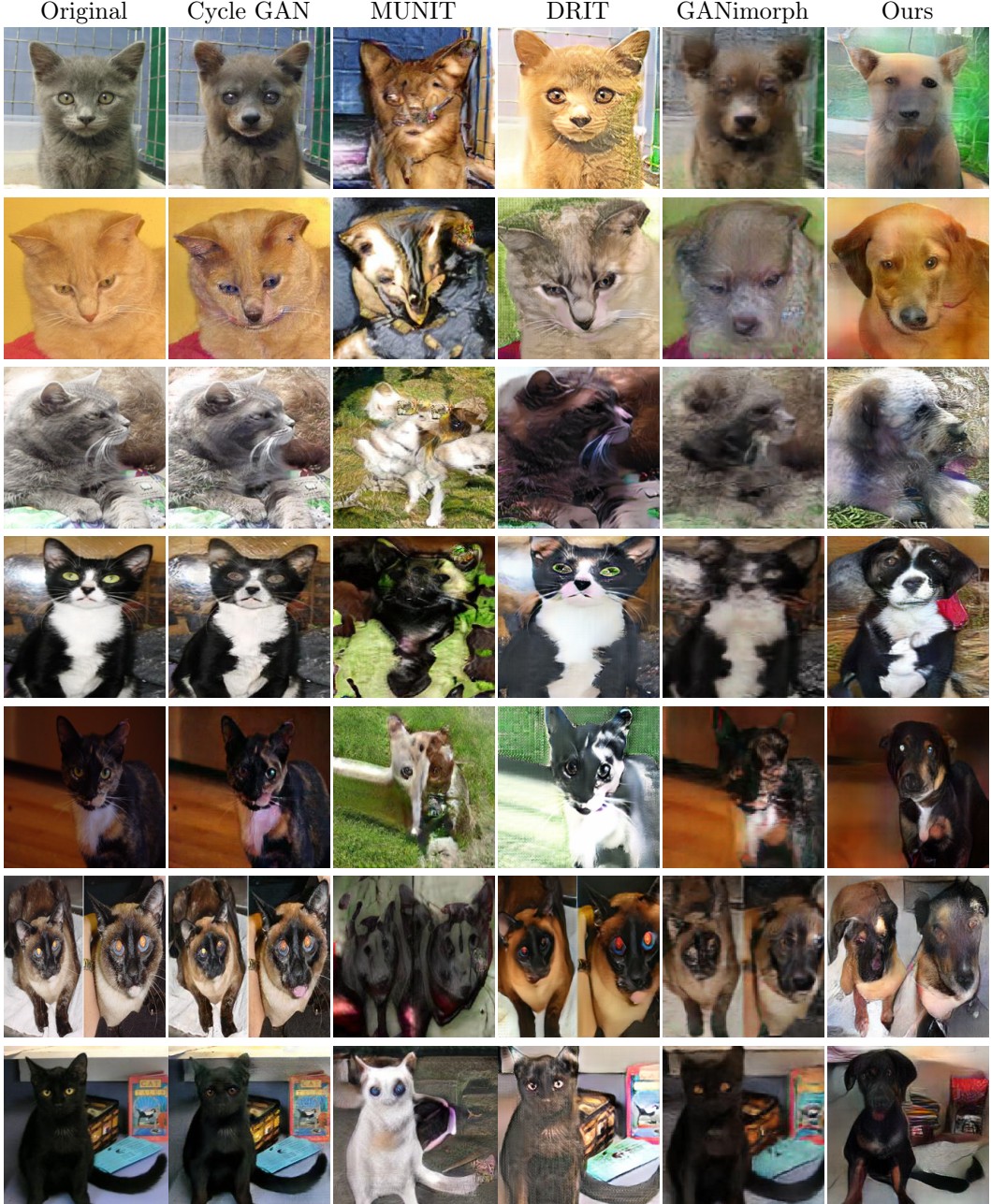

Figure 20: Qualitative comparisons. Kaggle cat to dog.

Original    Cycle GAN    MUNIT    DRIT    GANimorph    Ours

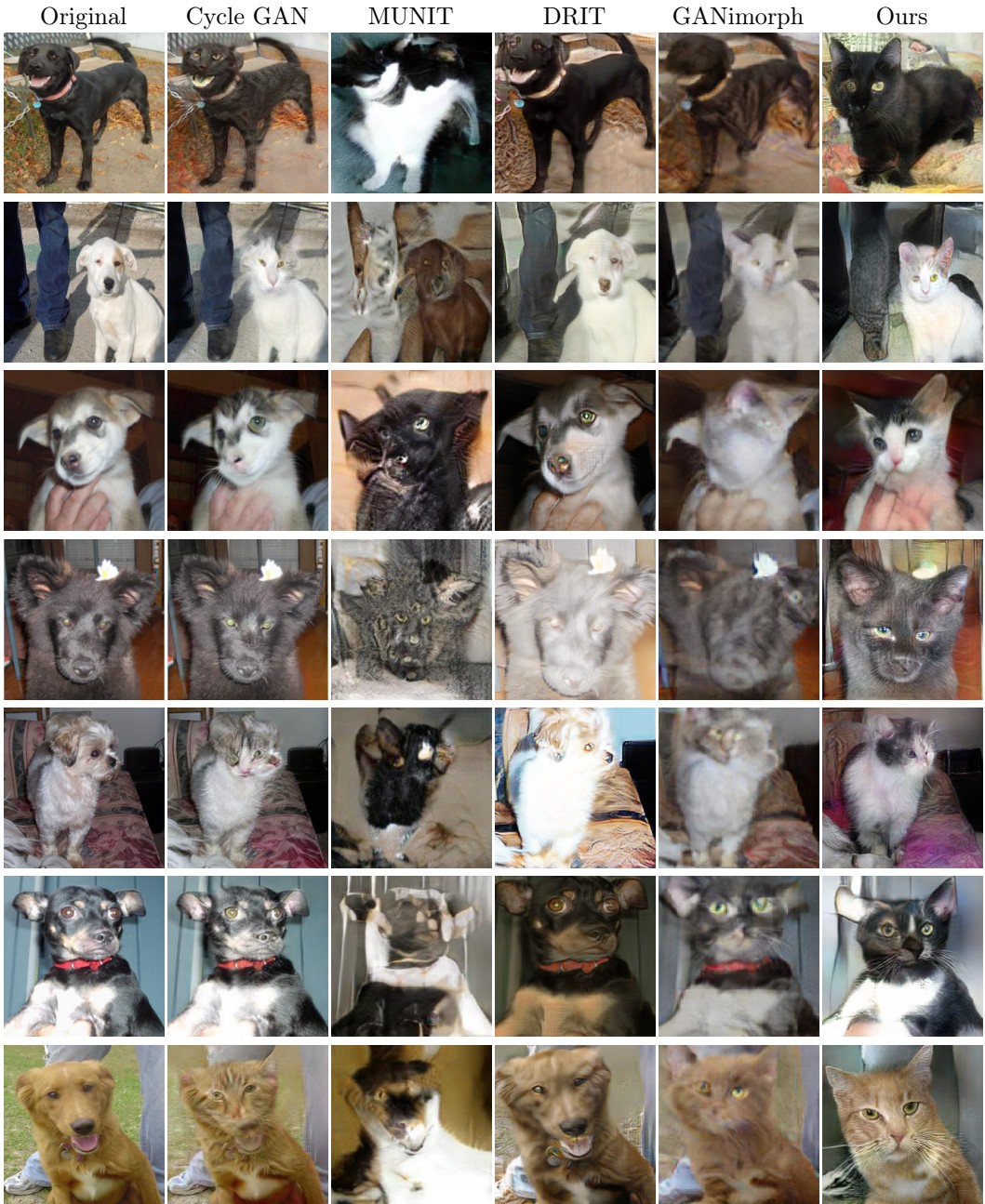

Figure 21: Qualitative comparisons. Kaggle dog to cat.

Original     Cycle GAN     MUNIT     DRIT     GANimorph     Ours

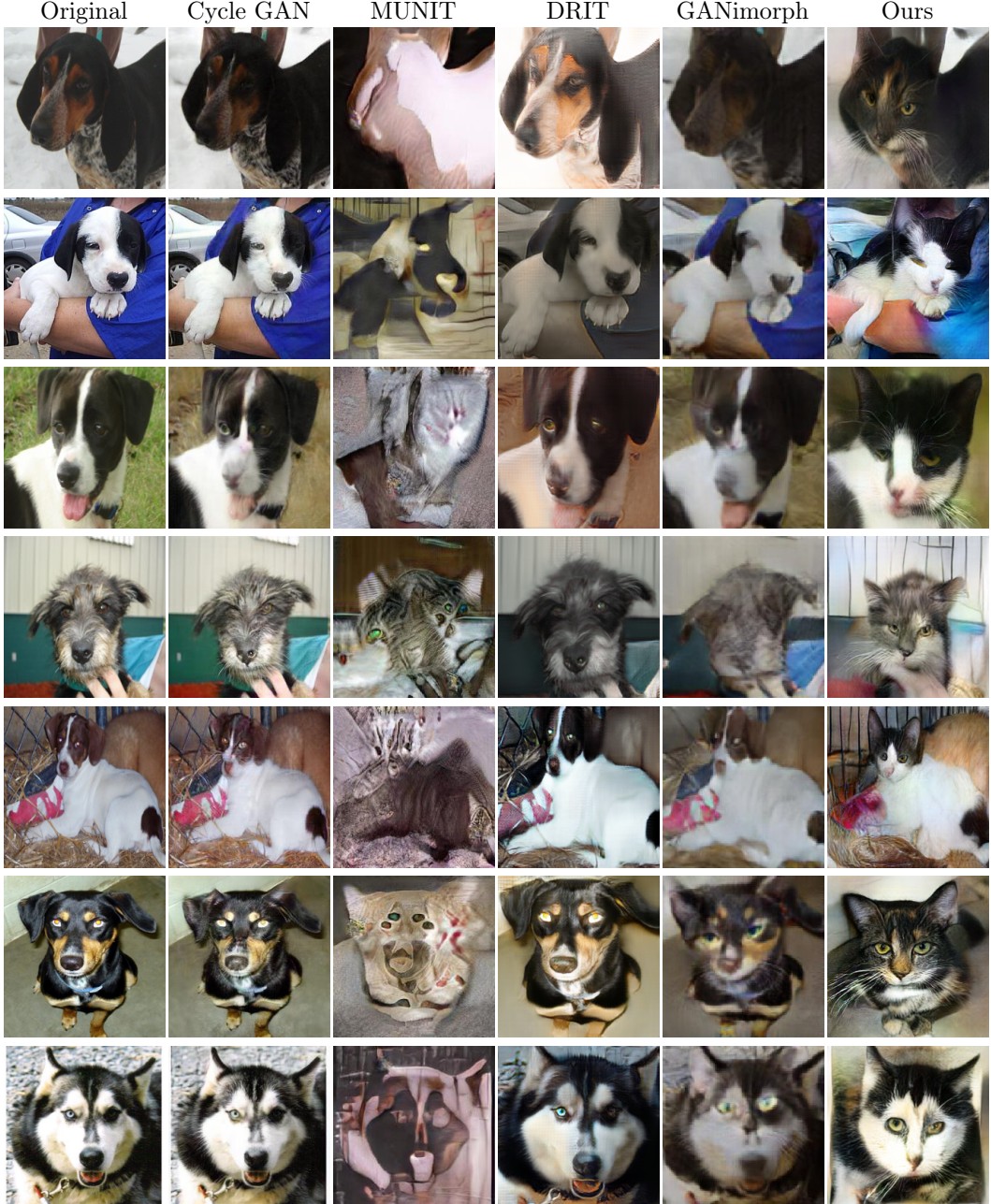

Figure 22: Qualitative comparisons. Kaggle dog to cat.

### A.3 NON-SHAPE DEFORMATION TRANSLATION

Our method is also suited to none-shape deformation tasks, as in the case of dataset(1) (Lee et al., 2018).

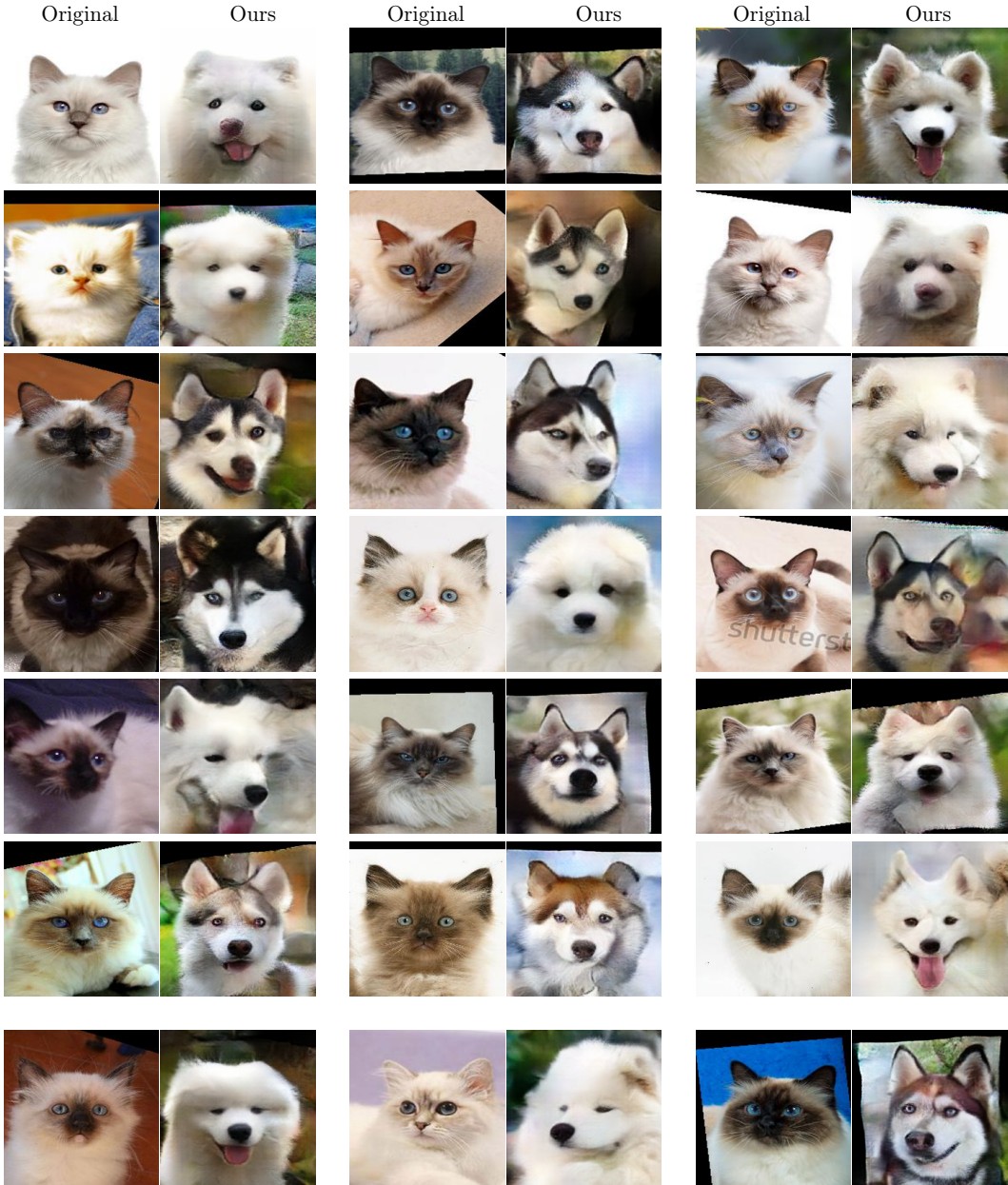

Figure 23: Translation results from cats to dogs (faces).

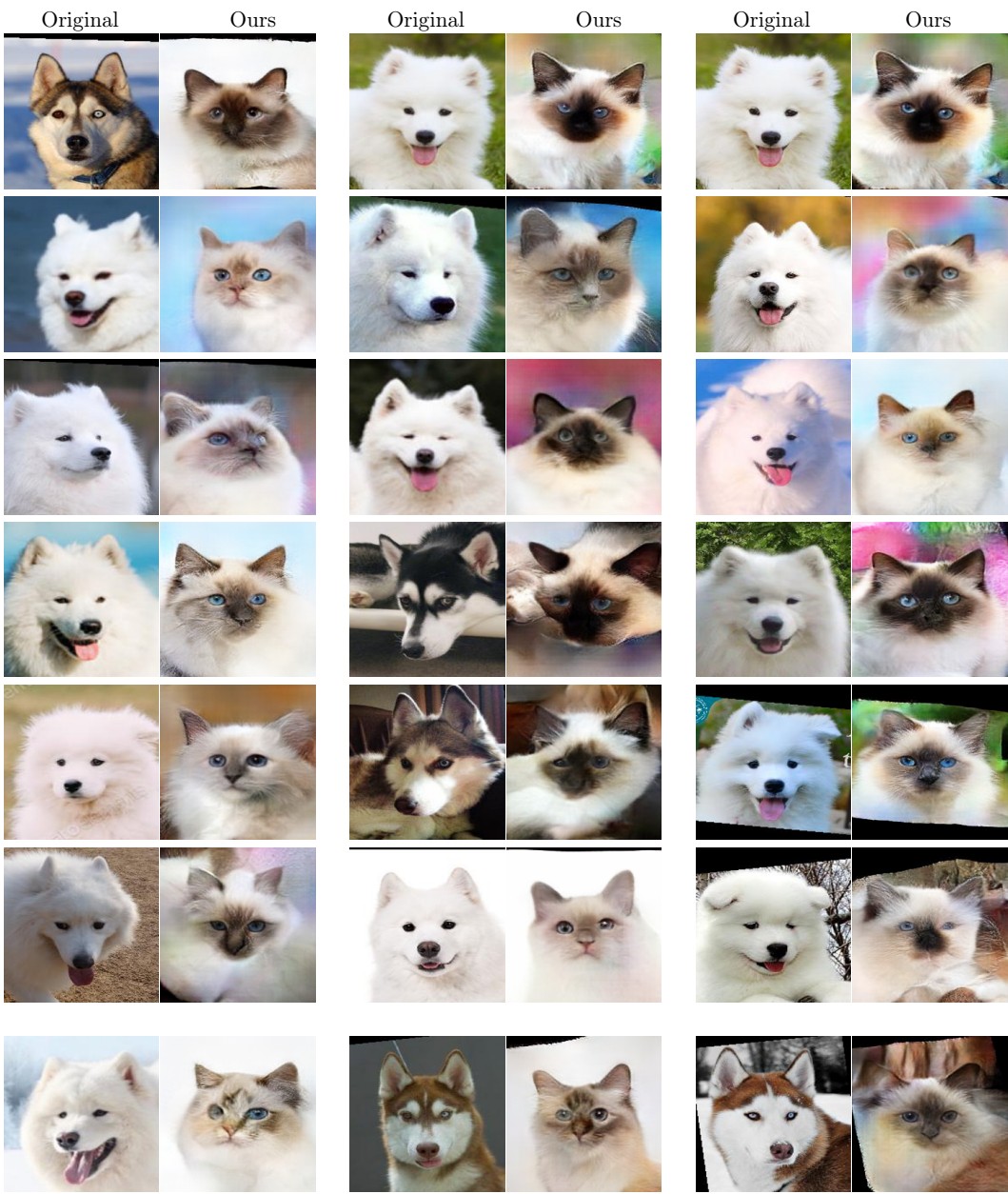

Figure 24: Translation results from dogs to cats (faces).

### A.4 COARSE TO FINE TRANSLATION

We here present the translation of each layer. The translation of each shallower layer is conditioned on the translation result of the previous layer, and learns to add fine scale and appearance, such as texture. At every layer, in order to visualize the generated deep features, we use a network pre-trained for inverting the deep features of VGG-19, following the method of Dosovitskiy & Brox (2016).

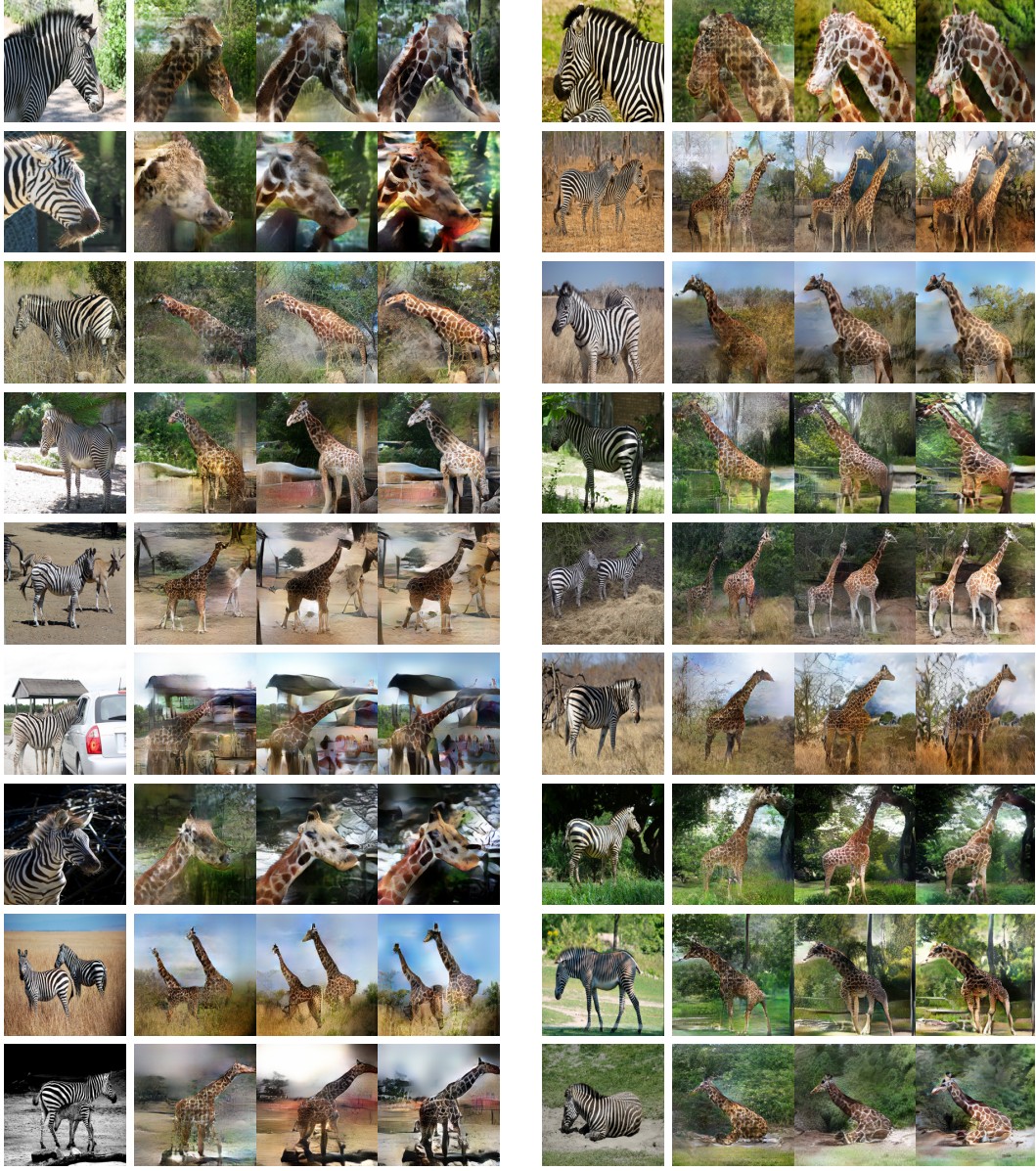

Figure 25: Coarse to fine translation of zebra to giraffe. Two different examples are shown in each row. The original image (left) is translated by the deepest translator (second left) and then in coarse to fine manner, shallower layers are translated (second right and most right).

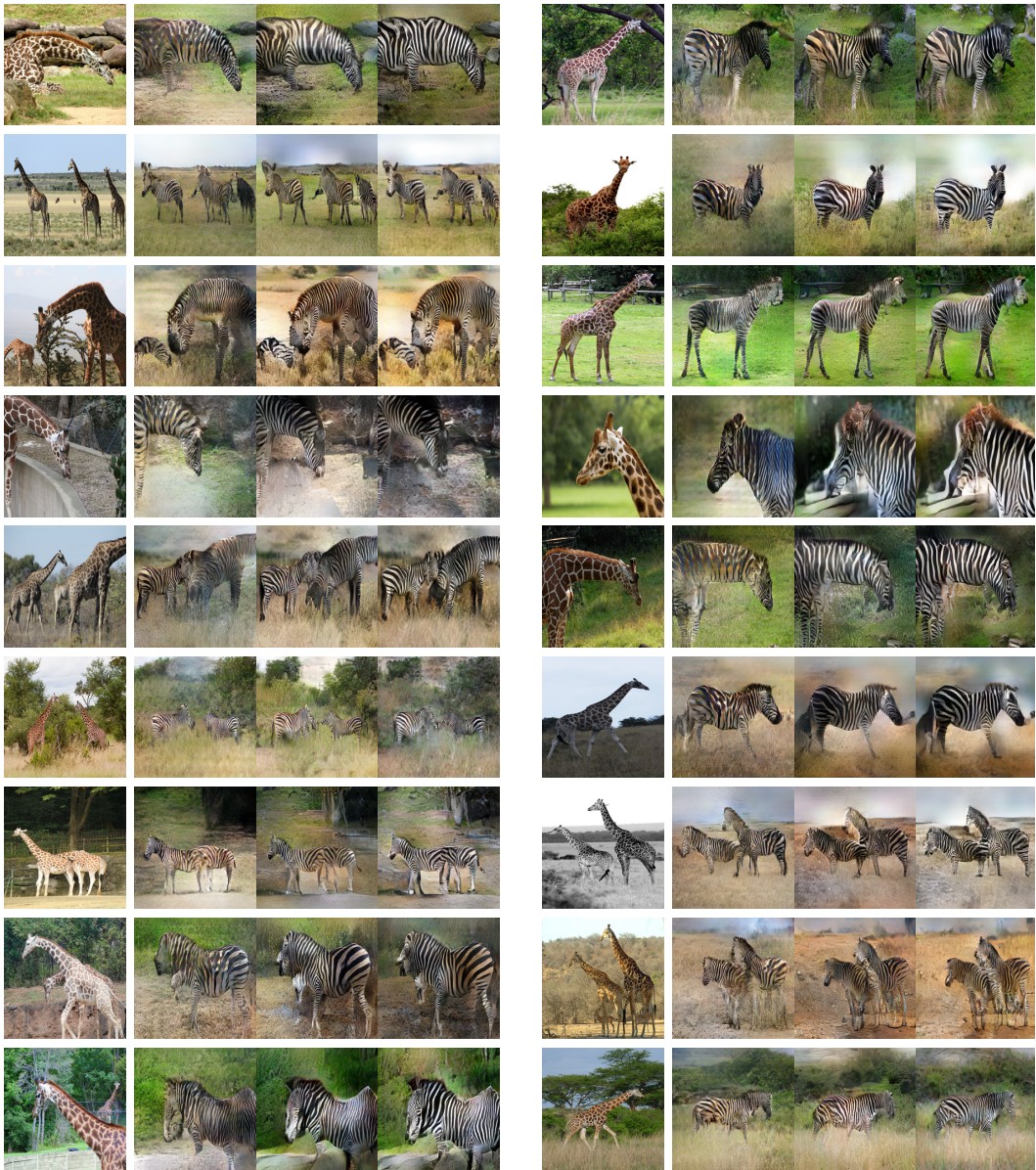

Figure 26: Coarse to fine translation of giraffe to zebra. Two different examples are shown in each row. The original image (left) is translated by the deepest translator (second left) and then in coarse to fine manner, shallower layers are translated (second right and most right).

## A.5    NEAREST NEIGHBOR COMPARISON

In this section we show side by side, source images, our translation and the three nearest neighbors in the target domain. We use the LPIPS metric, presented in Zhang et al. (2018). This metric is based on $L_2$ distance of deep features extracted from pre-trained network. In our case we use the default settings proposed by Zhang et al. (i.e. alex net). As we show, the closest image in the target dataset vary in pose, scale and content (i.e. different parts of the objects).

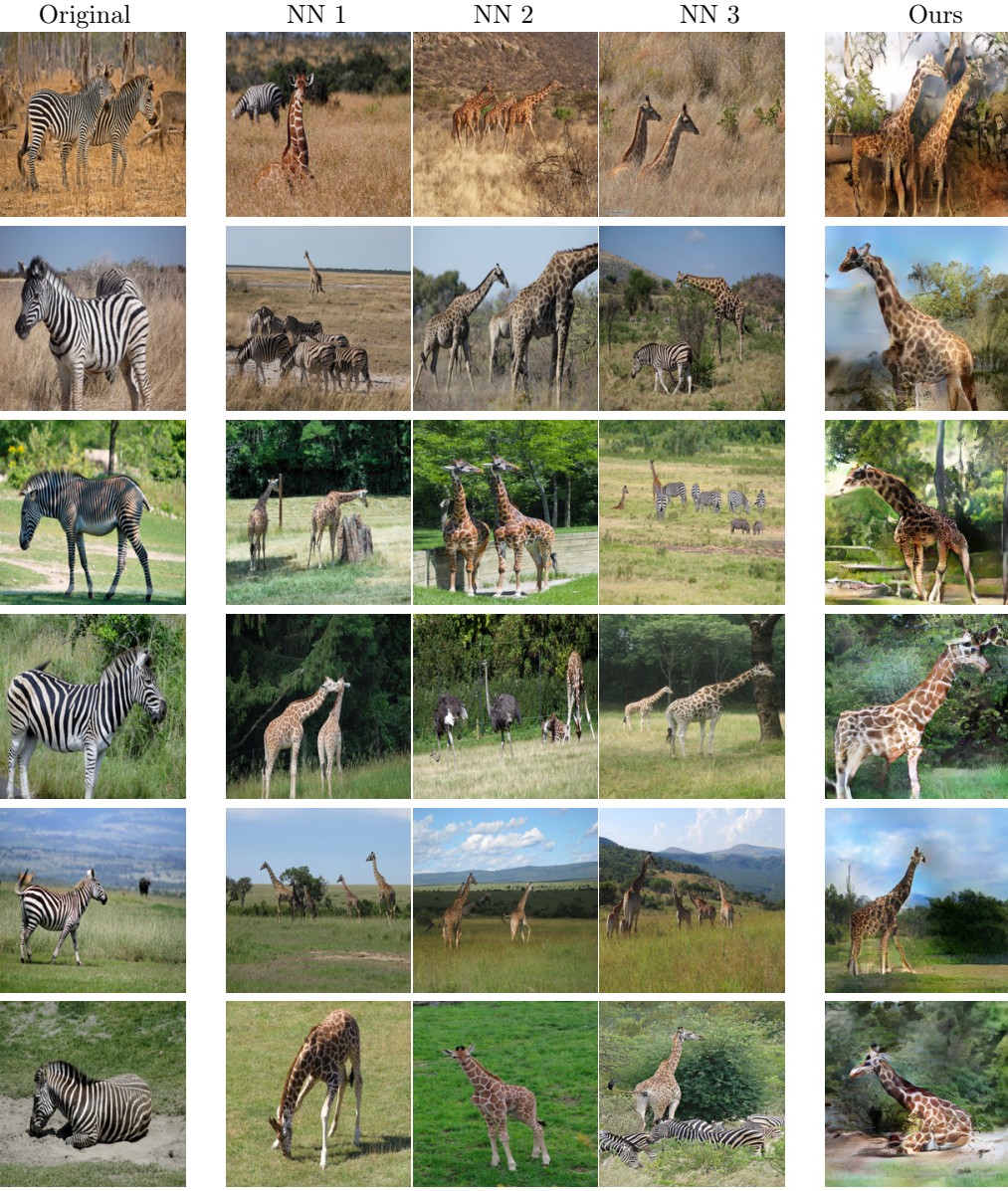

Figure 27: Nearest neighbor comparison to our result for zebra to giraffe translation. The NNs were found by exhaustive search on all the giraffe dataset using perceptual metric (LPIPS). The closest giraffe to the source zebra vary in scale, position and content

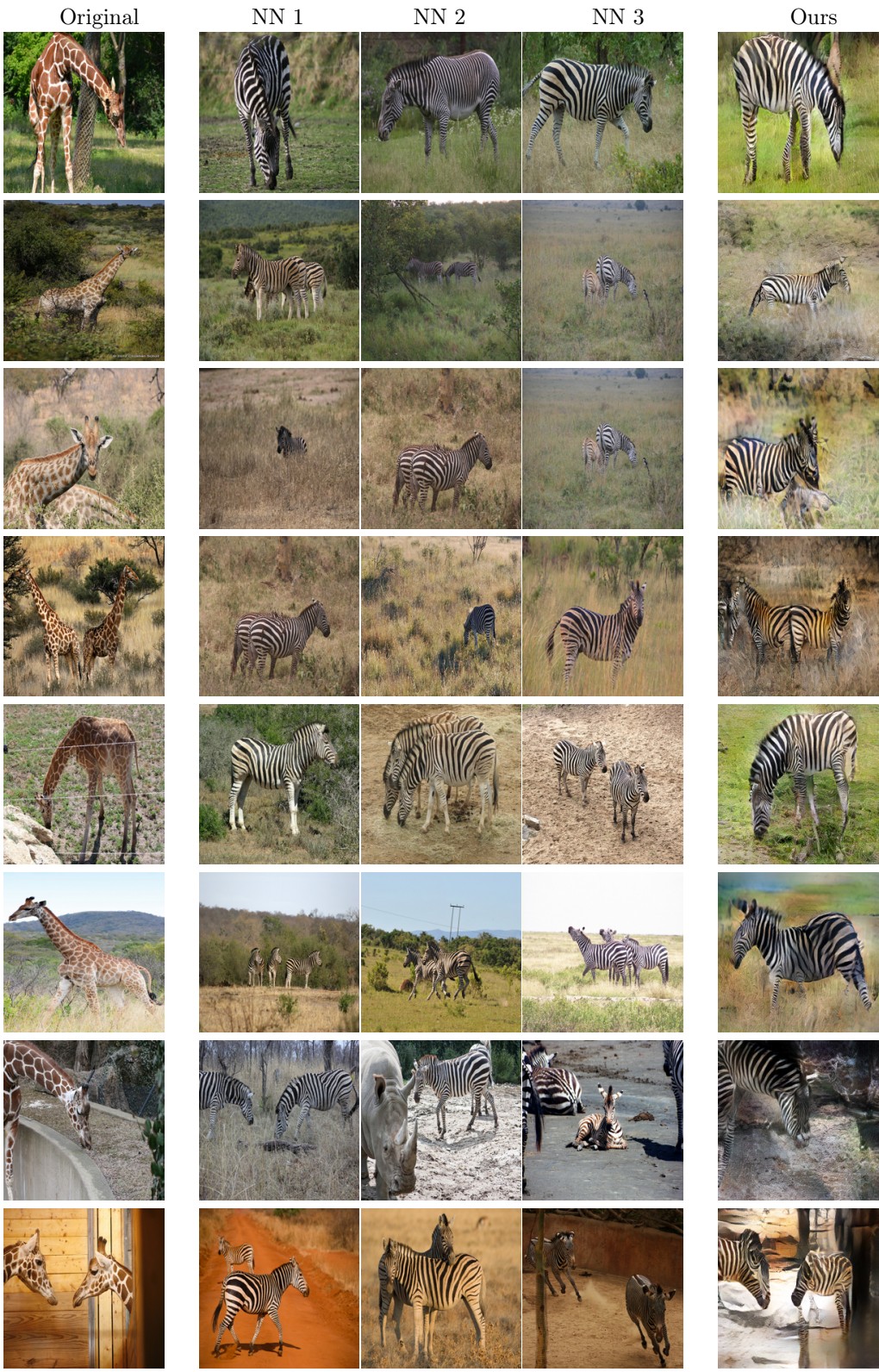

|  Original | NN 1 | NN 2 | NN 3 | Ours |
|---|---|---|---|---|

Figure 28: Nearest neighbor comparison to our result for giraffe to zebra translation. The NNs were found by exhaustive search on all the giraffe dataset using perceptual metric (LPIPS). The closest zebra to the source giraffe vary in scale, position and content.

