# OpenReview forum: "CROSS-DOMAIN CASCADED DEEP TRANSLATION"
_ICLR.cc/2020/Conference — Reject_

### Official Review · AnonReviewer1 · 2019-10-20
**Official Blind Review #1**

**Rating:** 6

**Review:**

The paper proposes a new method for image-to-image translation. The problem of most existing methods is that they are good in translating style  (e.g from photo to Van Gogh) but do not allow for significant changes in shape (e.g. from zebra to giraffe). The authors address this by performing the translation in a cascaded fashion starting from a semantic (deep) level (fifth layer of VGG). The underlying idea is that translating at this more semantic level puts less spatial constraints on the final resulting images (making translations from zebra to giraffe possible). After the fifth layer is translated other layers are translated conditioned on the previous translation results.

The method is compared to other translation methods including DRIT, MUNIT, GANimorph. Both visually and quantitatively as measured by FID. The results of the proposed method are superior. No comparison to TransGaGa is provided (but I could not find code for this method).

My recommendation is borderline accept. The proposed method is simple. The experimental results are limited but show both visually and quantitatively superior results. Especially FID scores are much better. I think the paper could be improved in explaining the conceptual novelty of the paper (especially with respect to GANimorph and TransGaGa).

1. I like the idea of applying the cycle consistency to the deeper layers rather than at the pixel level. Are there other methods which  do this ? It could be highlighted more as part of the contribution.
2. An ablation study should be added (in FID scores). I would like to see the necessity of the cascade (which is in the title) confirmed: results for only translating a single layer (3,4, or 5)  should be compared to translating 3,4,5 together as in proposed method.
3. Would it be possible to not use pretrained feature from VGG-19 ? This might also be a limitation. In principle, I guess you could train everything end-to-end, or is this impractical because of the feature inversion.
4.The authors could add some text on the lack of diversity for the translations in the limitation section. I understand there is no diversity and the translation is deterministic.

In general, I think the paper clearly explains what it does, and it also shows cases that it performs better than state-of-the-art. The paper could be much improved in its analysis of the reasons for its better performance, analyzing key aspects of its design like cycle GAN on features, pretrained VGG features and the use of cascaded generation of the final image.


**Experience Assessment:**

I have published one or two papers in this area.

**Review Assessment: Checking Correctness Of Derivations And Theory:**

N/A

**Review Assessment: Checking Correctness Of Experiments:**

I assessed the sensibility of the experiments.

**Review Assessment: Thoroughness In Paper Reading:**

I read the paper at least twice and used my best judgement in assessing the paper.

---

> ### Author Response · Authors · 2019-11-08
> **Response to Reviewer #1**
>
> Thank you for taking the time to review our paper and for your thoughtful suggestions and questions.
>
> --- cycle consistency to the deeper layers rather than at the pixel level. Are there other methods which do this ?
> We are not aware of any image to image translation methods, translating deep features directly or applying cycle consistency on pertained features. For completeness, as we point out in the related work, LOGAN (Yin et al.) pretrain an autoencoder, for point clouds, to create an overcomplete latent space. Then, translation is achieved between point clouds encoded by vectors in this latent space.
> However, the deep features that we use, are known to capture higher level semantics, as they were extensively trained for classification tasks. The key contribution of our work is to leverage the power of these deep features for the UNIT task.
>
> --- No comparison to TransGaGa is provided
> Unfortunately, the authors of TransGaGa are unable to publicly release their code, thus we are unable to compare our method to them. However, as acknowledged by the author of TransGaGa, both in the Appendix and in a private communication, it is robust to the single foreground and somewhat simple background. But, when the datasets are wild, busy and collected in unconstrained environments, TransGaGa fails.
>
> --- I think the paper could be improved in explaining the conceptual novelty of the paper (especially with respect to GANimorph and TransGaGa).
> We will try to explain this point better.
> Our conceptual novelty, can be viewed as a transfer learning for image translation as we are translating high level semantics, encoded in the deeper layers of a pre-trained classification network, a.k.a deep features. This is in contrast to existing UNIT methods, which learn to translate the images directly. GANImorph introduces architectural changes to cycleGAN, enabling higher deformation, and, as we show, we outperform this approach. TransGaGa, on the other hand, assumes intra and inter geometry consistency across the domains. This enables TransGAGA to disentangle geometry from appearance, in an unsupervised manner, and translate both separately (but this only works in limited scenarios, as explained earlier).
>
> --- An ablation study should be added (in FID scores) for a single layer (3,4, or 5).
> We will add such an ablation in our revision.
>
> --- Would it be possible to not use pertained feature from VGG-19 ?
> VGG19 is well suited for feature extraction as it gradually reduces the spatial dimension of the input while at the same time increasing the channel dimension. It has been shown in many previous works, that VGG19 extracts meaningful high level semantics, useful for style transfer, image analogy, etc. However, for the sake of completeness we can test and report the ability to use other feature extraction networks, such as AlexNet or inception network.
>
> --- The authors could add some text on the lack of diversity for the translations in the limitation section.
> Yes, the results are deterministic, similar to cycleGAN, this is indeed a limitation currently, common to UNIT methods. It would be indeed interesting to investigate how stochasticity may be added, as in many to many image translation methods, perhaps even to each layer translation separately.

---

### Official Review · AnonReviewer3 · 2019-10-23
**Official Blind Review #3**

**Rating:** 6

**Review:**

This paper proposes a new cascaded image-to-image translation method to address the I2I tasks where the domains have exhibit significantly different shapes. The proposed method train cycle GAN on different levels of feature extracted by pre-trained VGG and combine the futures with the AdaIN layer to keep the correct shape from the deep features.

Pros:
1. The proposed method seems to work well on different shape I2I datasets without using semantic masks compared to previous works.
2. The idea of cascaded translators sounds simple and reasonable which can probably benefit other related tasks. The way of applying AdaIn to combine features of different levels is also a nice trick to keep the correct shape from deep features.
3. The paper writing is OK, but some explanation and organization should be improved as mention in cons.

Cons:
1. Some figures are hard to understand without looking at the text. For example, in Figure 1, the caption does not explain the figure well. What does each image, the order, and the different sizes mean?  As to Figure 3, the words “top left image”, “right purple arrows” are a bit confusing.
2. The “Coarse to fine conditional translation” section describes the conditional translation in the shallow layers. I suggest mentioning it in previous sections for easy understanding.
3. As to the t-SNE visualization in Figure 9, different methods seem to use different N-D to 2-D mapping functions. This may lead to an unfair comparison.

Suggestions:
1. The authors use the pre-trained classification network VGG for feature extraction and then train dedicated translators based on these features. I wonder if the authors also tried finetuning VGG on the two domains or training an auto-encoder on the two domains. The domain-specific knowledge may help to improve the results and alleviate the limitations presented in the paper, e.g. background of the object is not preserved, missing small instances or parts of the object due to invertible VGG-19.

**Experience Assessment:**

I have published one or two papers in this area.

**Review Assessment: Checking Correctness Of Derivations And Theory:**

I carefully checked the derivations and theory.

**Review Assessment: Checking Correctness Of Experiments:**

I carefully checked the experiments.

**Review Assessment: Thoroughness In Paper Reading:**

I read the paper at least twice and used my best judgement in assessing the paper.

---

> ### Author Response · Authors · 2019-11-08
> **Response to Reviewer #3**
>
> Thank you for taking the time to review our paper and for your thoughtful suggestions and questions.
>
> --- Some figures are hard to understand without looking at the text
> ---The “Coarse to fine conditional translation”… I suggest mentioning it in previous sections for easy understanding.
> We will revise the captions of Figure1 and Figure3 to be more self-contained. The translation process is also explained in the introduction and at the beginning of the methods, and we will stress more the coarse to fine conditional translation.
>
> --- As to the t-SNE visualization in Figure 9, different methods seem to use different N-D to 2-D mapping functions. This may lead to an unfair comparison.
> As common for domain adaptation tasks, we calculate the t-SNE based on the source, target and translated features together.
>
> --- Finetuning VGG on the two domains or training an auto-encoder on the two domains.
> Fine-tuning VGG features is an interesting idea, which we did try for some of the datasets. However, we noticed this to produce slightly visually inferior results. This might be attributed to fixating on the exact differences between zebra and giraffe: scale, poses, and even the background.
> The use of an autoencoder, while enabling self-supervised semantics extraction, we believe, will struggle to achieve high quality semantics as successfully as VGG pertained on ImageNet. If requested, we could experiment with an autoencoder based on VGG architecture and report the results.

---

### Official Review · AnonReviewer2 · 2019-10-26
**Official Blind Review #2**

**Rating:** 3

**Review:**

* Summarize what the paper claims to do/contribute.
This paper claims to extend existing image translation works, like CycleGAN, to domain pairs that are not similar in shape. It is proposed to do so by using a VGG network trained on classification (I assume on Imagenet), extracting features from the two domains and learn 5 CycleGANs to translate for each level of the feature hierarchy. At each level of the hierarchy the translation from the previous level is used to condition the translation for the current level. During inference, the final image translation is done by "feature inversion" (a technique proposed in Dosovitsikiy and Brox, 2016) from the final feature layer. The technique is show on example from a number of pairs of domains like Zebra-to-Elephant (and back), Giraffe-to-Zebra (and back), Dog-to-Cat (and back) and is compared with a number of baselines qualitatively and quantitatively with the FID score.

* Clearly state your decision (accept or reject) with one or two key reasons for this choice.
Weak Reject.

Major reasons:
- The problem itself, as stated in the introduction, seems ill-posed to me. One of the struggles I had while looking through the results was to understand what the images should be looking like. ie What should a zebra translated to a giraffe look like? The motivation for such a problem is also not immediately clear either.
- Most of the resulting images do not seem "translated" to me. As stated in the paper (end of p.2) "one aims to transform a specific type of object without changing the background." As one can see in eg Fig. 1 the resulting translations are completely different images with the foreground object of the new domain in roughly similar poses. The background in most cases does not persist. What I suspect is actually happening here is that the high-level semantics from the first image are used as some sort of noise to generate new images from the new domain. One question I had, for example: could we be getting similar results if we used the VGG bottleneck as the noise vector in an InfoGAN? Since the VGG network is pretrained and used in the same way in both domains, I imagine we would be seeing something very similar. (and it would be def. preferrable to tuning 10 GANs!)

* Provide supporting arguments for the reasons for the decision.
Some of the decisions made in the paper were unclear and not supported adequately. The questions (in rough order of importance) that made some of the contributions unclear to me:
- Why wasn't a final translator used for the final image, conditioned on the final \tilde{b}_1?
- Is the VGG network pretrained on ImageNet? Why wasn't another task used that could be retaining more of the relevant features? eg on semantic segmentation
- Could this be used for networks pretrained on other datasets? Presumably ImageNet has information about the animals translated in this paper. Even better, could we somehow learn these features for the domain pairs automatically somehow?
- How meaningful is the FID score really in this case?
- How were the 10 GANs tuned?

* Provide additional feedback with the aim to improve the paper. Make it clear that these points are here to help, and not necessarily part of your decision assessment.
- It is mentioned on p.4 that "clamping is potentially a harmful irreversible operation" but that harmful results were not observed. As I was reading that I was wondering how these results would actually look like.
- On p. 6 it is mentioned that the number of images for 2 categories are reported in another paper. I think it'd take less space to actually report the number of images here.
- On p.7 it is mentioned that the number of instances is preserved, however it should be made clear that it's is perserved in some (or most if that is what was observed) of the examples.


**Experience Assessment:**

I have published one or two papers in this area.

**Review Assessment: Checking Correctness Of Derivations And Theory:**

I assessed the sensibility of the derivations and theory.

**Review Assessment: Checking Correctness Of Experiments:**

I carefully checked the experiments.

**Review Assessment: Thoroughness In Paper Reading:**

I read the paper at least twice and used my best judgement in assessing the paper.

---

> ### Author Response · Authors · 2019-11-08
> **Response to Reviewer #2**
>
> Thank you for taking the time to review our paper and for your thoughtful suggestions and questions.
>
> --- The problem itself is ill-posed.
> Generally UNIT (unpaired image translation) is an ill-posed task: what should a real image look like when translated to a Monet painting? One can imagine the outcome, yet there’s no precise definition. We would argue that the degree of ill-posedness depends on the domain. In the case of animal to animal translation, you would expect the result to contain a realistic looking animal with the same:
>     1. Semantic parts of an animal (i.e. head of a zebra to head of a giraffe). That also includes translating the correct amount of instances (i.e. 2 zebras to 2 giraffes).
>     2.  Location and scale of the objects (i.e. a small zebra at the left corner of the image should translate to small giraffe at the same location).
>     3. Pose
>     4. Background
> While we mostly succeed at 1-3, preserving the background is indeed problematic when translating deep features. However, this is common in UNIT. While shape non-deforming methods, such as cycleGAN, might not change the structures in the background, the color/style is typically changed. Shape deforming methods exhibit changes in both style and geometry of the background, see for example the recently proposed TransGaGa.
> While we made some preliminary attempts to incorporate an attention mechanism, the results were unsatisfactory, and we therefore stopped pursuing this direction, as we felt it to be outside the main focus of this work.
>
> --- One question I had, for example: could we be getting similar results if we used the VGG bottleneck as the noise vector in an InfoGAN?
> Using a different architecture instead of cycleGAN for unpaired deep feature translation is indeed interesting. Could the reviewer please elaborate exactly how did he envision here the use of infoGAN? Is the noise composed of a domain-part (i.e., zero or one with p=0.5 for each) and the VGG bottleneck features instead of the "traditional" noise?
> Regardless, directly inverting the bottleneck is difficult. We refer the reviewer to the results of the inversion network proposed by Dosovitskiy and Brox on AlexNet for different layers ("Generating Images with Perceptual Similarity Metrics based on Deep Networks"). In addition, as we show in our ablation study, the cascaded manner of our translation further improves the result achieved by the deepest layer translation only.
>
> ---Why wasn't a final translator used for the final image, conditioned on the final \tilde{b}_1?
> We noticed that shallow layers contribution was negligible. Thus, we omitted the use of \tilde{b}_1.
>
> --- Is the VGG network pretrained on ImageNet? Why wasn't another task used that could be retaining more of the relevant features? eg on semantic segmentation
> Yes, the VGG was pre-trained on ImageNet, we will clarify it in our revision.
> VGG pretrained on ImageNet is widely used for feature extraction, from perceptual similarity to cross domain correspondence. It is remarkable that a network pretrained only with image-level annotations can assist in the translation process.
> Semantic segmentation networks require more elaborate supervision (pixel-level annotation) and allow a different kind of translation approaches, which can directly use the segmentation maps.
>
>  --- Could this be used for networks pretrained on other datasets? Presumably ImageNet has information about the animals translated in this paper. Even better, could we somehow learn these features for the domain pairs automatically somehow?
>
> Yes, different networks can be used, as the approach is generic, although, a good feature extraction network, such as VGG pretrained on ImageNet, is required for a meaningful translation. Please note that while ImageNet does contain several of the animals translated in this paper, it does *not* contain giraffe and the different types of dogs and cats presented. We believe that learning the features in a self-supervised manner, will not yield the same quality as VGG features and fine-tuning VGG on the specific domains did not yield better results.
>
> --- How meaningful is the FID score really in this case?
> FID metric is still commonly used to assess how close fake and real samples are.
> FID uses layer of inception trained on ImageNet, thus, it is closely related to our deep features translation. In a sense, we minimize it directly.
>
>  --- How were the 10 GANs tuned
> The GANs were tuned manually, experimenting with several architecture (similar to all layers), and losses. Same parameters where used for all translation tasks. We found this process to be relatively simple.
>
> --- On p.7 it is mentioned that the number of instances is preserved, however, it should be made clear that it's preserved in some (or most if that is what was observed) of the examples.
> In most cases the number of instances was preserved, we will clarify it in our revision, thanks.

---

> > ### Author Response · Authors · 2019-11-08
> > **Additional comment**
> >
> > --- "clamping is potentially a harmful irreversible operation" but that harmful results were not observed.
> > We noted here that this operation is irreversible, thus, we should be careful when using it. However, for a specific domain, the inverter networks (from features to image) was able to overcome such clamping process at least visually. We can measure and report explicitly the reconstruction loss from features to image space, with and without clamping, though visually we did not notice such difference.

---

### Author Response · Authors · 2019-11-12
**Revision uploaded**

Dear reviewers,
We implemented several of your suggestions and comments, in a revised version of the paper (uploaded), and plan to continue to address the others.

We briefly summarize the changes in this revised version:
-  Figure 1 caption explained better, as well as modifications to Figure 3 caption.
-  Emphasis on our conceptual novelty w.r.t. other UNIT methods.
-  Additional ablation study:
      - FID comparison of different layers (layers 3,4,5) translations for 3 datasets in Table .1.
      - Qualitative comparison to a fine-tuned VGG network, for the zebra and giraffe dataset (Figure 9).
      - Qualitative comparison to AlexNet (pretrained on ImageNet), for the zebra and giraffe dataset (Figure 9).


We strongly believe that we propose a conceptually novel approach for large geometric deformations image-to-image translation.

If you have any further questions or suggestions, please do not hesitate to let us know.

Thank you again for your time and comments,
The Authors

---

### Decision · Program_Chairs · 2019-12-19

**Decision:**

Reject

**Comment:**

The paper addresses image translation by extending prior models, e.g. CycleGAN, to domain pairs that have significantly different shape variations. The main technical idea is to apply the translation directly on the deep feature maps (instead of on the pixel level).
While acknowledging that the proposed model is potentially useful, the reviewers raised several important concerns:
(1) ill-posed formulation of the problem and what is desirable, (2) using fine-tuned/pre-trained VGG features, (3) computational cost of the proposed approach, i.e. training a cascade of pairs of translators (one pair per layer).
AC can confirm that all three reviewers have read the author responses. AC suggests, in its current state the manuscript is not ready for a publication. We hope the reviews are useful for improving and revising the paper.